# SOLAX: A Python solver for fermionic quantum systems with neural network support

Louis Thirion[1], Philipp Hansmann[1], Pavlo Bilous[2*]

**1** Department of Physics, Friedrich-Alexander-Universität Erlangen-Nürnberg, 91058 Erlangen, Germany
**2** Max Planck Institute for the Science of Light, Staudtstraße 2, 91058 Erlangen, Germany
*pavlo.bilous@mpl.mpg.de

September 2, 2024

## Abstract

Numerical modeling of fermionic many-body quantum systems presents similar challenges across various research domains, necessitating universal tools, including state-of-the-art machine learning techniques. Here, we introduce SOLAX, a Python library designed to compute and analyze fermionic quantum systems using the formalism of second quantization. SOLAX provides a modular framework for constructing and manipulating basis sets, quantum states, and operators, facilitating the simulation of electronic structures and determining many-body quantum states in finite-size Hilbert spaces. The library integrates machine learning capabilities to mitigate the exponential growth of Hilbert space dimensions in large quantum clusters. The core low-level functionalities are implemented using the recently developed Python library JAX. Demonstrated through its application to the Single Impurity Anderson Model, SOLAX offers a flexible and powerful tool for researchers addressing the challenges of many-body quantum systems across a broad spectrum of fields, including atomic physics, quantum chemistry, and condensed matter physics.

# 1    Introduction

Accurate numerical modeling of fermionic quantum many-body systems presents an essential challenge across many research domains. In atomic physics, for example, precise knowledge of electronic energy levels is indispensable for the development of atomic frequency standards, the understanding of astrophysical spectra, and the search for phenomena beyond the Standard Model [1]. In particular, the promising yet scarcely explored domain of highly charged ions lacks experimental data and requires extensive computational support [2]. In quantum chemistry, the pursuit of highly accurate electronic

structure calculations, such as those achieved through full configuration interaction (full CI) methods [3–7], is crucial for the accurate determination of molecular properties and the prediction of chemical reactivity. In condensed matter physics, quantitative research on low-energy effective Hamiltonians such as the Hubbard model [8] and its derivatives, supports the qualitative understanding of microscopic mechanisms underlying phenomena like unconventional superconductivity in cuprates [9, 10], iron pnictides [11], and nickelates [12, 13]. While the core motivations and goals in all these diverse research areas are often completely different, the computational challenges are similar and the most challenging tasks are often identical from the technical point of view. Alongside methodological developments, advanced simulation codes for quantum many-body systems are therefore essential for scientific progress across a broad spectrum of fields.

However, many challenges still demand computational efforts that exceed the capabilities of even the most efficient codes and/or the available computational resources. In such cases, machine learning techniques can be applied to reduce the complexity of the calculation without compromising the accuracy. Possible approaches based on a neural network (NN) were demonstrated in Refs. [14, 15] for problems in computational atomic physics, and in Ref. [16] in a more general context of fermionic systems requiring large expansions of the wave function in the basis of Slater determinants. In the latter work, the machine learning functionality based on the TensorFlow library [17] was interfaced with the Quanty CI code [18]. Following the successful "proof of principle" in Ref. [16], we saw the need to implement an integrated Python library rather than interfacing existing codes. Here we present the resulting SOLAX package.

SOLAX is a comprehensive NN-boosted Python library designed for the study of fermionic quantum many-body systems. Within the standard quantum many-body formalism of second quantization, SOLAX provides a framework for constructing and solving quantum cluster problems. The SOLAX package offers a versatile set of tools to efficiently encode and manipulate basis sets, quantum states, and operators, enabling users to simulate and explore the structure of quantum many-body systems. The library supports the accurate determination of many-body quantum states in finite-size Hilbert spaces. Beyond its core functionalities, it includes built-in machine learning tools. Specifically, SOLAX addresses the exponential growth of Hilbert space dimensions in large quantum clusters: When full diagonalization becomes computationally infeasible, an NN classifier can be employed to approximate the solution through efficient basis optimization. The SOLAX library has already been successfully applied to the study of molecules such as $N_2$ [19]. The NN algorithm presented in this article was first demonstrated in Ref. [16] in its application to the Single Impurity Anderson Model (SIAM).

Importantly, SOLAX is structured in a modular way, allowing users to flexibly adapt and extend its components to meet specific research needs. The core low-level functionalities are implemented using powerful tools based on the Python library JAX recently developed at Google [20]. JAX offers highly efficient GPU-accelerated mechanisms to manipulate data organized as arrays similar to those from the well-known NumPy library [21]. This allows for flawless transition of data from and to the NumPy format, which is the main way to store data in SOLAX.

We assume the reader of this article to be familiar with the Python programming language and the libraries NumPy [21] and SciPy [22] which are both extensively used in scientific programming. The necessary information on the tools from the JAX ecosystem will be provided as they are used. We further assume that the reader is familiar with basic machine learning concepts using NNs. For an introduction to NN concepts, we refer to the classical work [23]. Machine learning from the general (probabilistic) perspective is discussed in depth in the comprehensive source [24, 25]. For a practical introduction

to machine learning (including neural networks using TensorFlow), we recommend the book [26].

The article is structured as follows. In Section 2 we demonstrate the core functionality of SOLAX for solving the fermionic many-body problem and show an exemplary computation for SIAM. In Section 3 we present built-in SOLAX tools for NN-supported computations. In Section 4 we discuss the mechanisms for saving/loading SOLAX objects and reproducing SOLAX computations. The article closes with conclusions in Section 5.

## 1.1 Code availability and dependencies

The SOLAX code can be cloned directly from the GitHub repository [27], where we also provide Jupyter notebooks with the code snippets shown in this article. Apart from packages from the standard Python library available without separate installation, SOLAX employs the following third-party libraries: NumPy [21], Pandas [28], SciPy [22], JAX [20], FLAX [29], and Orbax [30] (the latter 3 libraries belong to the JAX ecosystem). The versions of Python and the listed packages used in SOLAX at the moment of the present publication are summarized in Table 1. The user is required to perform the necessary installations before using SOLAX. Note that for leveraging GPU acceleration, a GPU-capable version of JAX must be installed.

| Python | NumPy | Pandas | SciPy | JAX | FLAX | Orbax |
|--------|-------|--------|-------|------|------|-------|
| 3.10.9 | 1.26.1 | 1.5.3 | 1.10.0 | 0.4.30 | 0.8.5 | 0.1.9 |

Table 1: The versions of Python and the third-party libraries used in SOLAX at the moment of the present publication.

## 2 Solver for fermionic quantum systems

The core functionality of the SOLAX package consists of encoding and solving eigenvalue equations for fermionic quantum many-body systems. Fully antisymmetric Slater determinants serve as the basis for many-body Hilbert spaces of fermionic wave functions. The occupation number representation of a Slater determinant is a binary string of zeroes "0" and ones "1" (adhering to the Pauli exclusion principle). As the full set of Slater determinants forms a complete basis on a given Hilbert space, any many-body quantum state in this space can be expanded using this basis. SOLAX follows this paradigm in the representation of quantum states. Operators within the SOLAX package are expressed in terms of creation $(\hat{a}_i^\dagger)$ and annihilation $(\hat{a}_i)$ operators which act on basis Slater determinants or quantum states represented as linear combinations of Slater determinants. These ladder operators are indexed by single-particle quantum numbers $i$ which indicate the positions in the occupation-number string on which they act. In this formalism, a standard Hamilton operator can be decomposed into single-particle, two-particle, and more generally, $n$-particle operators. Each of these terms in the Hamiltonian is expressed as a sum of ladder-operator products containing the corresponding number of creation and annihilation operators.

In the following, we describe how Slater determinant bases, many-body quantum states, operators, and their matrix representation can be defined and manipulated in SOLAX. We first introduce the main components implemented in SOLAX as Python classes: `Basis`, `State`, `OperatorTerm`, `Operator`, and `OperatorMatrix`. Subsequently, we demonstrate

these fundamental tools with the help of an example by finding the ground state of the paradigmatic Single Impurity Anderson Model (SIAM). The data storage and processing for the presented classes are primarily based on NumPy arrays [21], which also facilitate convenient interaction with the user. Internally, SOLAX enhances operations on 2D NumPy arrays using the Pandas library [28]. This powerful data analysis tool allows us to leverage hash maps and significantly speed up operations in which search in 2D NumPy arrays is needed. At the same time, the central quantum solver operation of acting with operators on quantum states is implemented in JAX [20] and supports automatic GPU acceleration.

## 2.1 Basis

We begin by introducing the `Basis` class in SOLAX, which facilitates efficient manipulation of basis sets composed of Slater determinants. To utilize SOLAX and NumPy in a Python environment, the libraries are imported as follows:

```python
import solax as sx
import numpy as np
```

### 2.1.1 Object construction

A `Basis` instance is created from a collection of strings, each representing a Slater determinant in the occupation number representation. All determinants (strings) must have the same length corresponding to the number of the underlying single-particle degrees of freedom which we refer to from here on as "spin-orbitals". For example, a `Basis` object `basis` with all possible Slater determinants for two electrons occupying four spin-orbitals is constructed as follows.

```python
basis = sx.Basis(["1100", "1010", "1001", "0110", "0101", "0011"])
```

SOLAX treats the received strings as binary representations of 8-bit integer numbers (in general a few for each determinant) and stores these efficiently in a 2D NumPy array. The length of each Slater determinant (i.e. the total number of 0s and 1s) in a `Basis` object can be accessed using the read-only `bitlen` property:

```python
print(basis.bitlen)
```

```
4
```

As will be seen in the following, `Basis` objects behave similarly to Python sets in many contexts. In particular, if repeated determinants are passed to the class constructor, these are automatically discarded. (We note that this automatic operation can be switched to manual control, which, however, goes beyond basic SOLAX usage and is not considered here.)

### 2.1.2 Conversion to a Python string and printing

The user can convert a `Basis` to a Python string and print it to see the stored determinants in the usual format:

```
print(basis)
```

```
1100
1010
1001
0110
0101
...
```

Note that only the first five (the default value of the print limit) determinants are reflected in the printed string. The value of this limit can be changed using the context manager `dets_printing_limit`:

```
with sx.dets_printing_limit(10):
    print(basis)
```

```
1100
1010
1001
0110
0101
0011
```

The user provides here the maximal number of printed determinants (in this case 10) or `None` if no limit is to be used.

### 2.1.3   Length, indexing, and slicing

**A. Basis as a Python sequence.**   The `Basis` class implements the Python sequence protocol, i.e. its objects have length and can be indexed and sliced in the standard way. The length is the number of the contained Slater determinants, and can be obtained with the Python built-in `len` function:

```
print(len(basis))
```

```
6
```

Indexing and slicing picks the Slater determinants at the corresponding positions and returns a new `Basis` object:

```
print(basis[0])
```

```
1100
```

```
print(basis[0:3])
```

```
1100
1010
1001
```

Note that in contrast to the typical behavior of sequences, `basis[0]` is again of type `Basis`. However, this is natural for this particular class and still complies with the Python sequence interface.

**B. "Fancy" and boolean indexing.**   On top of the standard Python sequence functionality, the `Basis` class supports the NumPy-style "fancy" and boolean indexing [31]. That is, `Basis` objects can be indexed in two additional ways: using a list on indices and with a boolean mask, respectively

```
print(basis[0, 1, 4])
```

```
1100
1010
0101
```

```
print(basis[True, True, False, False, True, False])
```

```
1100
1010
0101
```

As for NumPy arrays, the boolean mask here must have the same length as the sequence itself.

**Important Note!**   There is a crucial difference between the indexing mechanisms shown in (A) and (B), which is inherited from NumPy. In NumPy, standard Python indexing and slicing do not create a new array but instead reference a sub-array of the original array. In contrast, fancy or boolean indexing does create a new array in memory. This behavior extends to the `Basis` class, as it is based on NumPy (like most classes in SOLAX). While a new `Basis` object is created regardless, this does not necessarily hold for the underlying NumPy arrays. Therefore, for efficient memory usage, the indexing method described in (A) should be preferred.

### 2.1.4   Set operations

From an operational perspective, `Basis` objects behave similarly to Python sets. However, they do not fully implement the standard Python set interface. The reason for this is twofold: (1) not all set operations are necessary for SOLAX applications, and (2) the standard Python set nomenclature can be somewhat confusing in this context. Below, we describe the operations with `Basis` objects as implemented in SOLAX.

**A. Equality relation.**   As for Python sets, the equality relation `==` disregards the order of the Slater determinants in the compared `Basis` instances. For example, we compare the created `basis` object with its reverse:

```
print(basis == basis[::-1])
```

```
True
```

**B. Addition (set union).** Objects of the `Basis` class can be added (unified like sets) using the `+` operator. Note that the analogous operator for Python sets is `|`. As an example, consider two `Basis` objects obtained from `basis` by selecting only Slater determinants at even and odd positions, respectively:

```
basis_even = basis[::2]
basis_odd = basis[1::2]
```

As expected, their addition equals the initial `basis`:

```
print(basis_even + basis_odd == basis)
```

```
True
```

In the basic SOLAX usage, the resulting `Basis` has no repeated determinants. For example, unification of `basis` with itself gives again `basis` since no new determinants are added:

```
print(basis + basis == basis)
```

```
True
```

**C. Set difference.** `Basis` objects can be subtracted like sets. This operation is bound to the operator `%` instead of `-` as for Python sets. For instance, for the objects `basis`, `basis_even` and `basis_odd` introduced above, we have:

```
print(basis % basis_even == basis_odd)
```

```
True
```

Note that both for the `+` and the `%` operator, the Slater determinants in both operands must have the same length in the sense of the total number of 0s and 1s (which can be accessed using the `bitlen` property).

## 2.2 State

We proceed with the `State` class in SOLAX. While the `Basis` class incorporates sets of Slater determinants, the `State` class also includes real or complex coefficients assigned to each determinant. As such, a `State` object represents a quantum state expanded in the basis of Slater determinants. SOLAX supports linear algebra operations on `State` objects including their scalar product, thereby implementing the structure of a Hilbert space.

### 2.2.1 Fundamentals

Here we demonstrate the basic usage of the `State` class. As will be seen, many aspects are similar to those of the `Basis` class. A `State` instance is created from a `Basis` instance and a NumPy array of associated coefficients:

```
state = sx.State(basis, np.ones(6))
```

Here we used the `basis` object created in the previous section and a NumPy array with real numbers all equal 1. Note that states as represented by the SOLAX `State` class can be unnormalized, e.g. as the just created `state` object. Objects of the `State` class can be converted to a Python string and printed:

```
print(state)
```

```
|1100>  *  1.0
|1010>  *  1.0
|1001>  *  1.0
|0110>  *  1.0
|0101>  *  1.0
...
```

As for the `Basis` class, the number of shown determinants can be changed using the `dets_printing_limit` context manager. The underlying `Basis` object and the array of coefficients can be accessed directly as the attributes `basis` and `coeffs`:

```
print(state.basis)
```

```
1100
1010
1001
0110
0101
...
```

```
print(state.coeffs)
```

```
[1. 1. 1. 1. 1. 1.]
```

Analogously to the `Basis` class demonstrated in the previous section, the `State` class implements the Python sequence protocol and supports NumPy-style "fancy" and boolean indexing [31]. From the quantum mechanical perspective, this functionality corresponds to projecting the state onto the Hilbert subspace spanned by the selected determinants.

Additionally, SOLAX supports operations of the form `State % Basis`, which remove from the `State` object all determinants present in the `Basis` together with their associated coefficients. Quantum mechanically, this operation corresponds to projecting the state along the Hilbert subspace spanned by the deleted determinants. We demonstrate this operation using the `basis_even` and `basis_odd` objects from the previous section. The following code line retains in the resulting `State` only the determinants present in `basis_odd`:

```
state_odd = state % basis_even
print(state_odd)
```

```
|1010>  *  1.0
|0110>  *  1.0
|0011>  *  1.0
```

```
print(state_odd.basis == basis_odd)
```

```
True
```

The equality operator `==` is not directly implemented for the `State` class because the latter involves real or complex coefficients that are represented up to machine precision. Treating these accurately is important to avoid unexpected behavior. Later in this section, we will demonstrate a method for comparing `State` objects with user-defined accuracy.

### 2.2.2   Hilbert space operations

The `State` class represents quantum many-body states and supports corresponding operations in the Hilbert space.

**A. Multiplication by a scalar.**   A `State` object can be multiplied by a real or complex number resulting in a new `State` instance:

```
print(2 * state)
```

```
|1100>  *   2.0
|1010>  *   2.0
|1001>  *   2.0
|0110>  *   2.0
|0101>  *   2.0
...
```

The resulting `State` object contains a newly created NumPy array of `coeffs`, while the underlying `basis` is shared between the new and original `State` objects. The scalar can be used as both the left and right operand. In addition to multiplication by a scalar, SOLAX also supports division by a scalar and the unary `-` operator.

**B. Addition.**   Two `State` objects can be added using the `+` operator, which also applies the `+` operator to their underlying `Basis` objects, as described in the previous section. The coefficients associated with the same Slater determinants in the operands are summed. For instance, consider two `State` objects obtained from `state` via "fancy" indexing:

```
state1 = state[0, 1, 4]
print(state1)
```

```
|1100>  *   1.0
|1010>  *   1.0
|0101>  *   1.0
```

```
state2 = state[0, 3]
print(state2)
```

```
|1100>  *   1.0
|0110>  *   1.0
```

The addition operation gives:

```
print(state1 + state2)
```

```
|1100>  *   2.0
|1010>  *   1.0
|0101>  *   1.0
|0110>  *   1.0
```

The subtraction operation based on addition and unary negation is also supported:

```
print(state1 - state2)
```

```
|1100>  *   0.0
|1010>  *   1.0
|0101>  *   1.0
|0110>  *   -1.0
```

**Important Note!** As demonstrated in the previous example, Slater determinants with zero coefficients are not automatically removed in SOLAX. Instead, determinants with exact zero coefficients, as well as those with very small coefficients, must be manually removed using a user-defined cutoff. This approach allows us to treat equivalently the following examples of "zeros" (where only the first is exactly zero as represented in the machine):

```
print(0.1 + 0.1 - 0.2)
```

```
0.0
```

```
print(0.1 + 0.2 - 0.3)
```

```
5.551115123125783e-17
```

We show in the following how to "chop off" such zeros using SOLAX tools.

**C. Scalar product and normalization.** SOLAX supports Hermitian scalar product of `State` objects:

```
print(state1 * state2)
```

```
1.0
```

which allows to compute the norm as

```
print(state * state)
```

```
6.0
```

Once the norm is known, a `State` can be normalized using division by a scalar. SOLAX offers a shortcut for this operation implemented as the `normalize` method:

```
state_normalized = state.normalize()
print(state_normalized * state_normalized)
```

```
1.0000000000000002
```

The `normalize` method does not transform the initial `State` object but returns a new one. The underlying `basis` is shared.

### 2.2.3 Chopping and equality of states

Chopping a `State` object to a specified threshold is a useful operation implemented in SOLAX. As demonstrated below, the chop operation also enables the comparison of two `State` objects with a user-defined error margin.

**A. The chop method.** The `chop` method removes all Slater determinants from a `State` object whose coefficients have magnitudes smaller than a specified threshold. This operation creates a new `State` instance, leaving the original one unmodified. For demonstration, we consider the following `State`:

```
state3 = state2[-1]
state123 = state1 - state2 - state3
print(state123)
```

```
|1100>  *   0.0
|1010>  *   1.0
|0101>  *   1.0
|0110>  *  -2.0
```

Below we show 3 `State` objects obtained by chopping `state123` with respect to different thresholds:

```
state_chopped1 = state123.chop(1e-14)
print(state_chopped1)
```

```
|1010>  *   1.0
|0101>  *   1.0
|0110>  *  -2.0
```

```
state_chopped2 = state123.chop(1.5)
print(state_chopped2)
```

```
|0110>  *  -2.0
```

```
state_chopped3 = state123.chop(2.5)
print(state_chopped3)
print(len(state_chopped3))
```

```
0
```

As shown by the first example, the `chop` method allows the user to manually delete determinants with zero coefficients (we stress again that in SOLAX this is not done automatically).

**B. Equality of two State objects.** As mentioned above, the direct equality relation `==` is not implemented for the `State` class due to the finite machine precision of the involved real or complex coefficients. Instead, we can construct the difference of the `State` objects and chop the result with respect to a small cutoff. If the obtained `State` is empty, then the compared `State` objects were close within the precision determined by the threshold. For demonstration, we use the following `State` objects `state_a` and `state_b` which are not exactly equal due to the machine error of the involved coefficients:

```
state_mini = state[:2]
state_a = 0.1 * state_mini + 0.2 * state_mini
print(state_a)
```

```
|1100>  *   0.30000000000000004
|1010>  *   0.30000000000000004
```

```
state_b = 0.3 * state_mini
print(state_b)
```

```
|1100>  *   0.3
|1010>  *   0.3
```

Following the described procedure, we obtain:

```
state_diff = state_a – state_b
print(state_diff)
```

```
|1100>  *   5.551115123125783e-17
|1010>  *   5.551115123125783e-17
```

Now, chopping `state_diff` with respect to a small threshold gives an empty state:

```
state_zero = state_diff.chop(1e-14)
print(len(state_zero))
```

```
0
```

meaning that `state_a` and `state_b` are indeed equal within the error of `1e-14`. Practice shows that proper control here can be crucial to avoid unexpected behavior.

## 2.3 OperatorTerm

We now introduce quantum mechanical operators represented in SOLAX by two classes: `OperatorTerm` and `Operator`. The `OperatorTerm` class efficiently represents products of ladder operators with the same structure, while the `Operator` class encapsulates multiple `OperatorTerm` objects with different structures in a single entity. We will further discuss quantum operators as implemented in SOLAX through a concrete example, beginning with the `OperatorTerm` class.

Once more we consider the case of two electrons occupying four spin-orbitals. In this example, the four slots with indices 0, 1, 2, and 3 correspond to the electronic states $\uparrow^{(1)}$, $\downarrow^{(1)}$, $\uparrow^{(2)}$, and $\downarrow^{(2)}$, respectively. Here, the arrow and superscript indicate the spin and orbital quantum numbers, respectively. We examine the following hopping operator:

$$\hat{V} = v \left( \hat{a}_0^\dagger \hat{a}_2 + \hat{a}_1^\dagger \hat{a}_3 \right) + \text{h.c.} \tag{1}$$

and assume $v = 1$. Note that $\hat{V}$ consists of ladder operator products of the same structure $\hat{a}_i^\dagger \hat{a}_j$ and, therefore, can be represented by one `OperatorTerm` object. We start with the non-Hermitian operator

$$\hat{V}_0 = \hat{a}_0^\dagger \hat{a}_2 + \hat{a}_1^\dagger \hat{a}_3 \tag{2}$$

and gradually build up $\hat{V}$ using the functionality provided in SOLAX.

### 2.3.1 Object construction

Here we show how `OperatorTerm` objects are instantiated by considering the example of the introduced $\hat{V}_0$ operator. However, to keep the explanation generic, we denote the number of multipliers in each ladder operator product as $L$, and the number of the summed products as $K$ (specifically for $V_0$ we have $L = 2$ and $K = 2$).

Each `OperatorTerm` has 3 ingredients:

- `daggers` is a tuple of 0s and 1s of length $L$ showing which ladder operators in the products are annihilation (0s) and which are creation (1s) operators;

- `posits` is a 2D NumPy array of shape $K \times L$ and integer type indicating at which positions (as counted from 0) the ladder operators in each product act;

- `coeffs` is a 1D NumPy array of length $K$ containing real or complex coefficients for each product.

For $V_0$:

```
daggers = (1, 0)

posits = np.array([
    [0, 2],
    [1, 3]
])

coeffs = np.array([
    1.0,
    1.0
])
```

An `OperatorTerm` instance is then created as

```
V0 = sx.OperatorTerm(daggers, posits, coeffs)
print(V0)
```

```
OperatorTerm(
    daggers=(1, 0),
    posits=array([[0, 2],
            [1, 3]]),
    coeffs=array([1., 1.])
)
```

Unlike the `Basis` and `State` classes, which contain NumPy arrays with encoded Slater determinants that are not directly readable by the user, the `OperatorTerm` class displays its underlying NumPy arrays without any special formatting when converted to a Python string and printed. Additionally, if the `posits` array passed to `OperatorTerm` contains repeated rows, these duplicates are automatically removed, and the corresponding coefficients in `coeffs` are summed. A manual mode is also available for advanced usage not addressed in this article.

### 2.3.2   Similarities with the State class

The `OperatorTerm` class has strong similarities with the `State` class. Conceptually, they both represent expansions over some basis elements (ladder operator products and Slater determinants, respectively). At the technical level, both classes incorporate a 2D NumPy array with an accompanying 1D coefficient array. We list here the analogous features without going into comprehensive details.

- `OperatorTerm` implements the Python sequence protocol and supports the NumPy-style "fancy" and boolean indexing [31] (see also the section on the `Basis` class).

- Objects of `OperatorTerm` can be added using the `+` operator. If the `daggers` tuples of the summands are equal (i.e. the quantum operators have the same structure), the result is of the `OperatorTerm` type and otherwise of the `Operator` type (the latter class is considered in the next section).

- The `OperatorTerm` class supports multiplication with scalars.

- `OperatorTerm` objects can be "chopped" with respect to a real threshold using the `chop` method.

- Equality relation `==` is not implemented for the `OperatorTerm` class. As for the `State` class, the equality up to a user-determined precision can be checked with the binary `-` operator and subsequent chopping.

### 2.3.3   Hermitian conjugate

The `OperatorTerm` class supports the operation of Hermitian conjugation via the `hconj` property returning a new `OperatorTerm` object:

```
print(V0.hconj)
```

```
OperatorTerm(
    daggers=(1, 0),
    posits=array([[2, 0],
            [3, 1]]),
    coeffs=array([1., 1.])
)
```

Using the introduced functionality, we can construct now the full operator $\hat{V}$ as

```
V = V0 + V0.hconj
print(V)
```

```
OperatorTerm(
    daggers=(1, 0),
    posits=array([[0, 2],
            [1, 3],
            [2, 0],
            [3, 1]]),
    coeffs=array([1., 1., 1., 1.])
)
```

Note that the Hermitian conjugate $\hat{V}_0^\dagger$ has the same operator structure as $\hat{V}_0$, and hence the sum is represented by an `OperatorTerm` object.

### 2.3.4    Acting on states and bases

`OperatorTerm` objects represent quantum mechanical operators and can therefore act on quantum states. We demonstrate now how this functionality is implemented in SOLAX. Considering again the introduced example, we construct a normalized quantum state $|\Psi\rangle$ with the total spin $S_z = 0$:

```
basis = sx.Basis(["1001", "0110"])
coeffs = np.array([1.0, -1.0])

psi = sx.State(basis, coeffs)
psi = psi.normalize()

print(psi)
```

```
|1001>  *   0.7071067811865475
|0110>  *  -0.7071067811865475
```

The operator action $\hat{V}|\Psi\rangle$ can be now performed as a call

```
result_psi = V(psi)
print(result_psi)
```

```
|1100>  *   1.414213562373095
|0011>  *   1.414213562373095
```

Additionally, it is possible to act with an `OperatorTerm` directly on `Basis` objects. The result is then also of type `Basis` and contains the same Slater determinants as when acting on a `State`:

```
result_basis = V(psi.basis)
print(result_basis)
```

```
1100
0011
```

As will be demonstrated later in this work, the operation of acting directly on `Basis` objects is useful for iterative basis extension procedures via acting with operators.

### 2.3.5   GPU acceleration and batches

Acting with `OperatorTerm` is implemented using the JAX library [20] which supports computations on an NVIDIA GPU. Therefore, if such GPU is available on the machine and a GPU-capable version of JAX is installed, it will be automatically used for this operation.

**Batching.**   Since GPU memory is often scarce, we provide the user with the possibility to perform the operator action in batches by using the call arguments `det_batch_size` and `op_batch_size`. They are responsible for batching the `State` (or `Basis`) object and the `OperatorTerm` object, respectively. Note that these are keyword-only arguments and have to be provided with the argument names explicitly. For example:

```
result_psi_batches = V(psi, det_batch_size=1, op_batch_size=2)
```

We can now use the standard procedure to ensure that the `State` objects obtained with and without batching are equal within a very small error:

```
s = result_psi_batches - result_psi
s = s.chop(1e-14)
print(len(s))
```

```
0
```

Note, however, that the internal ordering of the Slater determinants in the resulting objects may be different as computed with and without batching. We note also that in this demonstration example, very small batch sizes are chosen, but in general, the user should aim at maximally exhausting the GPU memory.

**Multiple GPUs.**   Parallelization across multiple GPUs is currently under development and available for some SOLAX functions. Acting with an `OperatorTerm` can be switched to a multi-GPU mode by providing the keyword-only argument `multiple_devices=True`. Then, the batches of the `State` (or `Basis`) object will be automatically distributed over a all GPUs if available on the machine. Note that also the argument `det_batch_size` must be provided to have multiple such batches. We stress again that this functionality is currently under development and is partially available to the user as an experimental feature.

## 2.4 Operator

Thus far, we have considered quantum operators consisting of ladder operator products with the same structure, differing only in the positions on which the ladder operators act. These are represented in SOLAX using the `OperatorTerm` class. As the next step, we introduce the `Operator` class, which encapsulates `OperatorTerm` objects of different structures along with a scalar term. An `Operator` object can encode any quantum operator expressed in the second quantization formalism using annihilation and creation operators.

While computations can be performed using only the `OperatorTerm` class introduced in the previous section, we recommend the users to follow these guidelines:

- prefer the `Operator` class for basic usage like acting on quantum states or basis sets;

- access the underlying `OperatorTerm` objects to fine-tune the operator.

In this section we will demonstrate this approach in practice.

### 2.4.1 Construction of simple operators

The basic way to instantiate `Operator` objects is based on the same ingredients as for the `OperatorTerm` class, i.e. `daggers`, `posits` and `coeffs` (see the previous section). These arguments can be passed directly to the `Operator` constructor:

```
op = sx.Operator(daggers, posits, coeffs)
print(op)
```

```
Operator({
    (1, 0): OperatorTerm(
        daggers=(1, 0),
        posits=array([[0, 2],
                [1, 3]]),
        coeffs=array([1., 1.])
    )
})
```

**Explanation of the printing output.** From the technical perspective, the `Operator` class implements the Python mapping protocol. In practice, this means that its objects behave similarly to Python dictionaries. Specifically, `Operator` objects store their underlying `OperatorTerm` objects as values in key-value pairs, with the keys being the corresponding `daggers` tuples. It is also possible to include a scalar term, which is associated with the string key `"scalar"`. This structure is reflected when `Operator` objects are converted to a Python string and printed, as demonstrated in the example above.

Once `daggers`, `posits` and `coeffs` are received, SOLAX creates automatically an `OperatorTerm` from the provided arguments and wraps it in an `Operator` object for convenient usage and further extension. In particular, this can be seen from the printed output for the `op` object above. Alternatively, the same `Operator` can be instantiated directly from the `OperatorTerm`:

```
print(sx.Operator(V0))
```

```
Operator({
    (1, 0): OperatorTerm(
        daggers=(1, 0),
        posits=array([[0, 2],
                [1, 3]]),
        coeffs=array([1., 1.])
    )
})
```

Here we reused the `OperatorTerm` object `V0` created in the previous section.

### 2.4.2   Addition as a way to build operators

The `Operator` created above is still trivial in the sense that it hosts only one `OperatorTerm`. More advanced `Operator` objects can be constructed from more basic ones using addition. Also addition of an `Operator` with an `OperatorTerm` or a scalar leads to another `Operator`. Moreover, as mentioned in the previous section, addition of two incompatible `OperatorTerm` objects does not lead to an error but creation of an `Operator` hosting the summands. For illustration, we introduce the following on-site energy operator:

$$\hat{U} = u_{01}\, \hat{a}_0^\dagger \hat{a}_0 \hat{a}_1^\dagger \hat{a}_1 + u_{23}\, \hat{a}_2^\dagger \hat{a}_2 \hat{a}_3^\dagger \hat{a}_3 \tag{3}$$

with $u_{01} = 0.25$ and $u_{23} = 0.75$ as an example. We encode it as an `Operator` using the described standard way:

```
daggers_u = (1, 0, 1, 0)
posits_u = np.array([
    [0, 0, 1, 1],
    [2, 2, 3, 3]
])
coeffs_u = np.array([0.25, 0.75])

U = sx.Operator(daggers_u, posits_u, coeffs_u)
print(U)
```

```
Operator({
    (1, 0, 1, 0): OperatorTerm(
        daggers=(1, 0, 1, 0),
        posits=array([[0, 0, 1, 1],
                [2, 2, 3, 3]]),
        coeffs=array([0.25, 0.75])
    )
})
```

Now we can construct e.g. the compound operator

$$\hat{H} = \mathbb{1} + \hat{V} + \hat{U} = \mathbb{1} + \hat{V}_0 + \hat{V}_0^\dagger + \hat{U} \tag{4}$$

directly as

```
H = 1 + V0 + V0.hconj + U
print(H)
```

```
Operator({
    (1, 0): OperatorTerm(
        daggers=(1, 0),
        posits=array([[0, 2],
                [1, 3],
                [2, 0],
                [3, 1]]),
        coeffs=array([1., 1., 1., 1.])
    ),
    scalar: 1,
    (1, 0, 1, 0): OperatorTerm(
        daggers=(1, 0, 1, 0),
        posits=array([[0, 0, 1, 1],
                [2, 2, 3, 3]]),
        coeffs=array([0.25, 0.75])
    )
})
```

As seen from the printed output, the resulting `Operator` consists of the scalar 1 and two `OperatorTerm` objects.

### 2.4.3   Similarities with the OperatorTerm class

The `Operator` class contains operations which are similar to the `OperatorTerm` class.

- Linear operations of addition and multiplication with scalars.

- Hermitian conjugation using the `hconj` property.

- Action on `State` and `Basis` objects — is delegated to the underlying `OperatorTerm` components and multiplication with the scalar with subsequent addition of the partial results. Note that in case of action on a `Basis`, the scalar, if present, acts effectively as the unity operator (even if it is equal to zero).

- The keyword-only batching arguments `det_batch_size` and `op_batch_size` are available for the `Operator` action, and control batching for the `OperatorTerm` components.

- We remind the user here, that the underlying `OperatorTerm` objects automatically support computations on an NVIDIA GPU.

- For leveraging the multi-GPU parallelization (experimental!), the `Operator` action call can receive the keyword-only argument `multiple_devices=True` which propagates to the underlying `OperatorTerm` objects. In this case, batches corresponding to `det_batch_size` will be distributed over a few GPUs, if available.

### 2.4.4   Manipulations with Operator objects

**A. Accessing underlying components.**   A key difference between the `Operator` and `OperatorTerm` classes is the container type they implement. `OperatorTerm` objects are sequences and can be indexed using integer positions and slices, as well as NumPy's "fancy" and boolean indexing mechanisms. In contrast, the `Operator` class implements the Python mapping protocol, making it similar to Python dictionaries, which are indexed via their

keys. For `Operator` objects, the keys are either the `daggers` tuples or the `"scalar"` string. For example:

```python
print(H[1, 0, 1, 0])
```

```
OperatorTerm(
    daggers=(1, 0, 1, 0),
    posits=array([[0, 0, 1, 1],
           [2, 2, 3, 3]]),
    coeffs=array([0.25, 0.75])
)
```

```python
print(H["scalar"])
```

```
1
```

Note that for indexing, the "tuple" parenthesis for `daggers` can be omitted as in the example above. As usual Python dictionaries, `Operator` instances support views `keys`, `values` and `items` for iteration over their entries.

**B. Operator length.** `Operator` objects as mappings have length, which however only reflects the number of the stored components and is not related to the length of the underlying `OperatorTerm` objects. Indeed:

```python
print(len(H))
```

```
3
```

whereas for the contained `OperatorTerm` components the lengths are:

```python
for key, term in H.items():
    if key != "scalar":
        print(f"Length of the OperatorTerm {key} is {len(term)}")
```

```
Length of the OperatorTerm (1, 0) is 4
Length of the OperatorTerm (1, 0, 1, 0) is 2
```

**C. Dropping components.** We have seen how `Operator` instances can be enriched with new components using addition. Conversely, if there is a need to remove an `OperatorTerm` or the scalar, this can be done using the `drop` method, which takes the key of the component to be removed and returns a new `Operator` object without it. The original `Operator` remains unmodified. For example:

```python
H_without_V = H.drop(1, 0)
print((1, 0) in H_without_V)
```

```
False
```

```
H_without_scalar = H.drop("scalar")
print("scalar" in H_without_scalar)
```

```
False
```

Here we used the Python `in` keyword for membership checks. This becomes automatically possible since the `Operator` class implements the mapping interface.

**D. Chopping OperatorTerm components.**   The chopping operation is implemented for the `Operator` class as the `chop` method, but can be applied only to a particular `OperatorTerm` via its `daggers` key. The returned `Operator` object contains the chopped version of the corresponding `OperatorTerm` or does not contain it at all if it has become empty after chopping. The initial `Operator` stays unchanged. Chopping for the scalar term is not supported. For instance, chopping `U` with respect to the cutoff 0.5 is performed as

```
H_chopped1 = H.chop((1, 0, 1, 0), 0.5)
print(len(H_chopped1[1, 0, 1, 0]))
```

```
1
```

leaving only one entry out of the two in the corresponding `OperatorTerm`. Chopping with respect to 1.0 leads to chopping it off completely:

```
H_chopped2 = H.chop((1, 0, 1, 0), 1.0)
print((1, 0, 1, 0) in H_chopped2)
```

```
False
```

## 2.5   OperatorMatrix

To solve a quantum many-body eigenvalue problem, it is often necessary to construct the matrix representation of a quantum mechanical operator (e.g. Hamiltonian) on a given basis set. We address this requirement with the SOLAX class `OperatorMatrix`, which provides tools for efficient matrix construction. It is important to note that the `OperatorMatrix` class is designed solely for the efficient construction of the operator matrix, while subsequent diagonalization is performed by the user with the help of the SciPy library [22].

### 2.5.1   Obtaining an OperatorMatrix

For the demonstration, we reconsider again the case of two electrons in four spin-orbitals. We use the `Operator` object H constructed in the last section and create here also a `Basis` of Slater determinants with the spin projection $S_z = 0$:

```
basis = sx.Basis(["1001", "1100", "0110", "0011"])
```

The matrix of the operator $\hat{H}$ on this basis can be built directly using the `build_matrix` method of the corresponding `Operator` object (this method is available also for the `OperatorTerm` class):

```
matrix = H.build_matrix(basis)
```

The result is an object is of the `OperatorMatrix` class, which stores matrix elements in a coordinate sparse format, meaning only non-zero matrix elements are stored. Here, we highlight the main features of the matrix construction operation:

- It is possible to construct non-square matrices using two distinct `Basis` objects for rows and columns by passing them as arguments to the `build_matrix` method.

- `build_matrix` supports the keyword-only arguments `det_batch_size` and `op_batch_size` which are used in evaluation of the matrix elements via action of the `OperatorTerm` objects (see the section on the `OperatorTerm` class).

- Matrix evaluation inherits from the `OperatorTerm` class the possibility to perform computations automatically on an NVIDIA GPU, if available.

- Computations on multiple GPUs (experimental!) can be switched on by providing `multiple_devices=True` to the `build_matrix` method. In this case, batches corresponding to `det_batch_size` will be distributed over a few GPUs, if available (see the section on the `OperatorTerm` class).

The dimensions of the constructed matrix can be accessed using the read-only `size` property:

```
print(matrix.size)
```

```
(4, 4)
```

The number of the non-zero matrix elements (i.e. the total number of the stored matrix elements) can be obtained as

```
print(matrix.num_nonzero)
```

```
12
```

The content of an `OperatorMatrix` object can be viewed after conversion to SciPy and NumPy which we discuss in the following.

### 2.5.2 Conversion to SciPy and NumPy

The built `OperatorMatrix` can be now converted to the SciPy format using the `to_scipy` method:

```
coo_matrix = matrix.to_scipy()
```

The returned object is of type `scipy.sparse.coo_matrix`, and stores the matrix in the coordinate sparse format similarly to the `OperatorMatrix` class. If necessary, the user can now convert it to a different sparse format using the SciPy means. Here, we convert the matrix to the usual NumPy dense format and print it:

```
dense_matrix = coo_matrix.todense()
print(dense_matrix)
```

```
[[ 1.    1.    0.    1.  ]
 [ 1.    1.   -1.    0.  ]
 [ 0.   -1.    1.25 -1.  ]
 [ 1.    0.   -1.    1.  ]]
```

Note that the NumPy dense representation is feasible only for small matrices, e.g. for demonstration or testing purposes. In this section we will often perform such conversion of `OperatorMatrix` objects in order to print their content in a usual matrix format. Therefore, we define the shortcut function:

```python
def print_matrix(m):
    print(m.to_scipy().todense())
```

### 2.5.3 Manipulations with OperatorMatrix objects

Once obtained from an `Operator` or `OperatorTerm`, the `OperatorMatrix` object allows for further useful manipulations, which we present here. In all examples considered below, a new transformed `OperatorMatrix` instance is created while the original remains unmodified. Here, we always utilize the `matrix` object constructed and shown above.

**A. Displace.** The `displace` method allows to shift the matrix content along the row and the column axes with the corresponding change of the matrix dimensions. The two method arguments are the number of positions the matrix is displaced by vertically (row axis) and horizontally (column axis), respectively. The shifts can be positive and negative. For example:

```python
print_matrix(
    matrix.displace(2, 1)
)
```

```
[[ 0.    0.    0.    0.    0.  ]
 [ 0.    0.    0.    0.    0.  ]
 [ 0.    1.    1.    0.    1.  ]
 [ 0.    1.    1.   -1.    0.  ]
 [ 0.    0.   -1.    1.25 -1.  ]
 [ 0.    1.    0.   -1.    1.  ]]
```

```python
print_matrix(
    matrix.displace(-1, -1)
)
```

```
[[ 1.   -1.    0.  ]
 [-1.    1.25 -1.  ]
 [ 0.   -1.    1.  ]]
```

As seen from these examples, the newly created positions are effectively filled with zeros, whereas the entries with resulting negative positions are dropped.

**B. Window.** The `window` method implements a rectangular filter, which sets all elements outside this rectangle to zero without changing the matrix shape. Technically, the filtered out matrix elements are directly discarded, since only non-zero matrix elements are stored in `OperatorMatrix` objects. For instance:

```
print_matrix(
    matrix.window((1, 1), (3, 4))
)
```

```
[[ 0.    0.    0.    0.  ]
 [ 0.    1.   -1.    0.  ]
 [ 0.   -1.    1.25 -1.  ]
 [ 0.    0.    0.    0.  ]]
```

The tuples passed to the `window` method correspond to the positions of the left upper (inclusive) and the right lower (exclusive) corners of the filter. If tuples $(a, b)$ and $(c, d)$ are passed, then the matrix values at the intersection of rows $i$: $a \leq i < b$ and columns $j$: $c \leq j < d$ survive, whereas the other matrix elements become zero. If any of the arguments $a$, $b$, $c$, $d$ is `None`, it will be replaced by a position leading to the largest possible filter size.

**C. Shrink basis.** After the matrix on a particular basis is constructed, it is straightforward to obtain the matrix on any sub-basis by extracting the corresponding matrix elements. This can be done using the `shrink_basis` method as we show in the following. We choose the sub-basis of our `basis` object corresponding to the electrons occupying different spatial orbitals:

```
sub_basis = basis[0, 2]
print(sub_basis)
```

```
1001
0110
```

Then the matrix of the same operator $\hat{H}$ on this sub-basis can be obtained as

```
print_matrix(
    matrix.shrink_basis(basis, sub_basis)
)
```

```
[[1.    0.  ]
 [0.    1.25]]
```

By default, this operation shrinks the basis along both axes. It is also possible to shrink the basis only along the row or the column axis by passing an additional argument `axis=0` or `axis=1` to the `shrink_basis` method, respectively.

**Important note!** Both the sub-basis and the initial basis have to be passed to the `shrink_basis` method. Therefore, it is important to perform the sub-matrix extraction prior to any matrix displacements, since displaced matrices are not related to the initial basis anymore.

**D. Chopping.** The `chop` method provides the possibility to discard matrix elements with absolute values less than a user-defined threshold. These entries are effectively set to zero (we remind that zeros are not stored in `OperatorMatrix` objects). As an example, we chop our matrix with respect to the threshold `1.1`:

```
print_matrix(
    matrix.chop(1.1)
)
```

```
[[0.   0.   0.    0.  ]
 [0.   0.   0.    0.  ]
 [0.   0.   1.25 0.  ]
 [0.   0.   0.    0.  ]]
```

### 2.5.4 Linear operations and Hermitian conjugate

Like other SOLAX classes related to quantum mechanical operators, the `OperatorMatrix` class supports addition, multiplication by real or complex scalars, and Hermitian conjugation via the `hconj` property. However, it is important to note that the addition operation deviates from the standard linear algebra convention. Specifically, `OperatorMatrix` objects can be added regardless of their shape. If the dimensions do not match, the matrices are effectively padded with zeros to form a minimal rectangle that encompasses both matrices before being added. This behavior is inherited from and natural for the `OperatorMatrix` implementation in the sparse format. To demonstrate this feature, we create two matrices of size $3 \times 3$ and $5 \times 2$ from our initial $4 \times 4$ matrix using displacements and then add them to obtain a matrix of size $5 \times 3$:

```
matrix3_3 = matrix.displace(-1, -1)
print_matrix(matrix3_3)
```

```
[[ 1.25 -1.    0.  ]
 [-1.    1.   -1.  ]
 [ 0.   -1.    1.75]]
```

```
matrix5_2 = matrix.displace(1, -2)
print_matrix(matrix5_2)
```

```
[[ 0.    0.  ]
 [ 0.    1.  ]
 [-1.    0.  ]
 [ 1.   -1.  ]
 [-1.    1.75]]
```

```
print_matrix(
    matrix3_3 + matrix5_2
)
```

```
[[ 1.25 -1.    0.  ]
 [-1.    2.   -1.  ]
```

```
[-1.   -1.    1.75]
[ 1.   -1.    0.  ]
[-1.    1.75  0.  ]]
```

Later in this work we will use the introduced manipulations for efficient computations with operator matrices.

### 2.5.5   Equality of OperatorMatrix objects

As also the other SOLAX classes containing real or complex numbers, the `OperatorMatrix` class does not directly support the equality relation and the `==` operator. The user can check closeness of non-zero elements in two `OperatorMatrix` objects by

a) subtracting them;

b) using the `chop` method with respect to a small threshold;

c) ensuring that the `num_nonzeros` property returns zero.

Note that due to the specific `OperatorMatrix` implementation, this procedure disregards completely the matrix dimensions and only checks closeness of non-zero elements having the same position in the matrices. For example, consider the following matrices `m1` and `m2`:

```
m1 = matrix.window((0, 0), (2, 2))
print_matrix(m1)
```

```
[[1. 1. 0. 0.]
 [1. 1. 0. 0.]
 [0. 0. 0. 0.]
 [0. 0. 0. 0.]]
```

```
m2 = matrix.shrink_basis(basis, basis[:2])
print_matrix(m2)
```

```
[[1. 1.]
 [1. 1.]]
```

The outlined procedure gives

```
print(
    (m1 - m2).chop(1e-14).num_nonzero == 0
)
```

```
    True
```

That is, all non-zero matrix elements having the same position are equal (within the chosen accuracy), but still the dimensions may differ, as in the considered case. The latter can be additionally compared as

```
print(m1.size == m2.size)
```

```
    False
```

## 2.6 Demonstration computation for SIAM

Having introduced the foundational components of the SOLAX package, we now demonstrate its application to SIAM. This example illustrates how SOLAX can be used to construct and analyze quantum systems by efficiently handling basis sets, states and operators specifically to find the ground state of a complex quantum many-body model.

### 2.6.1 Model description

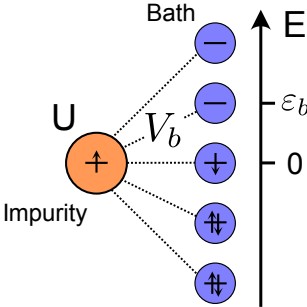

Figure 1: Schematic illustration of SIAM with $N_{\mathrm{bath}} = 5$ bath sites. The model is considered in the "star geometry", in which the correlated impurity (orange circle) hybridizes with non-interacting and uncorrelated bath sites (blue circles). The impurity energy is set to $\varepsilon_{\mathrm{imp}} = 0$. The on-site impurity interaction is given by $U$ [see Eq. (6)]. The bath site energies are $\varepsilon_b$ and the hybridization strengths are $V_b$.

We consider SIAM with the so called "star geometry" (see Fig. 1). We set the impurity energy to $\varepsilon_{\mathrm{imp}} = 0$ and denote the bath site energies as $\varepsilon_b$ and the hybridization strengths as $V_b$. Additionally, we include an on-site interaction for the impurity with strength $U$ [see Eq. (6)]. The full Hamiltonian is

$$\hat{H} = \hat{H}_{\mathrm{imp}} + \hat{H}_{\mathrm{bath}} + \hat{H}_{\mathrm{hyb}} \tag{5}$$

with the impurity term

$$\hat{H}_{\mathrm{imp}} = U \left( \hat{a}^{\dagger}_{\mathrm{imp},\uparrow} \hat{a}_{\mathrm{imp},\uparrow} - \frac{1}{2} \right) \left( \hat{a}^{\dagger}_{\mathrm{imp},\downarrow} \hat{a}_{\mathrm{imp},\downarrow} - \frac{1}{2} \right) , \tag{6}$$

the bath term

$$\hat{H}_{\mathrm{bath}} = \sum_{b=1}^{N_{\mathrm{bath}}} \varepsilon_b \sum_{\sigma \in \{\uparrow,\downarrow\}} \hat{a}^{\dagger}_{b,\sigma} \hat{a}_{b,\sigma} , \tag{7}$$

and the hybridization term

$$\hat{H}_{\mathrm{hyb}} = \sum_{b=1}^{N_{\mathrm{bath}}} V_b \sum_{\sigma \in \{\uparrow,\downarrow\}} \hat{a}^{\dagger}_{\mathrm{imp},\sigma} \hat{a}_{b,\sigma} + \mathrm{h.c.} \tag{8}$$

In all computations performed here, we consider an odd number of bath sites $N_{\mathrm{bath}}$ and assume that the system is half-filled with electrons. The model, as described here and exemplified for $N_{\mathrm{bath}} = 5$, is illustrated in Fig. 1. The energies of the bath sites $\varepsilon_b$ and the hybridization strengths $V_b$ are chosen similarly to Ref. [16] and are constructed using

the `build_bath` function based on the provided $N_{\text{bath}}$. As an example, Fig. 2 shows $\varepsilon_b$ and $V_b$ for $N_{\text{bath}} = 21$.

```python
def build_bath(N_bath):
    ii = np.arange(N_bath) + 1
    xx = ii * np.pi / (N_bath + 1)
    e_bath = -2 * np.cos(xx)

    V0 = np.sqrt(20 / (N_bath + 1))
    V_bath = V0 * np.sqrt(1 - (e_bath / 2)**2)

    return e_bath, V_bath
```

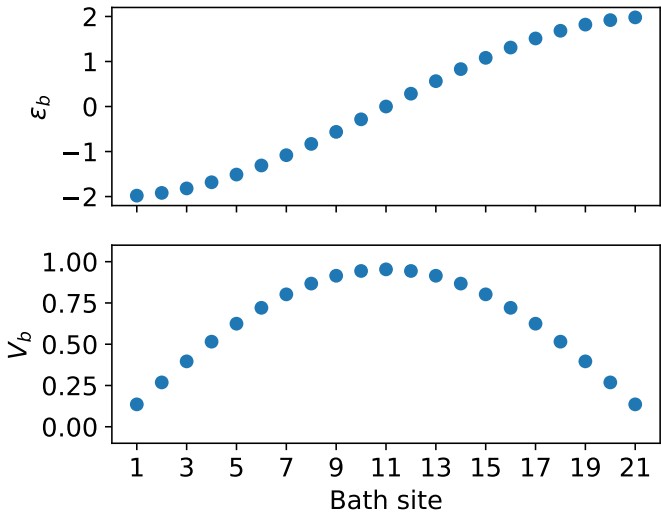

Figure 2: The bath site energies $\varepsilon_b$ and the hybridization strengths $V_b$ for $N_{\text{bath}} = 21$.

### 2.6.2   Eigenvalue problem and solution procedure

We aim to compute the energy of the ground state, which is known to belong to the $S_z = 0$ sector. An exact solution would require the construction and partial diagonalization of the Hamiltonian matrix over the basis set of all possible Slater determinants with $S_z = 0$. However, even with a few tens of bath sites, the complete basis becomes combinatorially large and numerically intractable. Therefore, it is common to perform these computations iteratively on a growing partial basis. Starting with a few Slater determinants representative of the target state, the following iterations are performed:

- construction and partial diagonalization of the Hamiltonian matrix on the current basis set;

- extension of the basis set by acting on it with an extension operator $\hat{O}$ (here we choose $\hat{O} = \hat{H}$).

The energies obtained in each iteration at the diagonalization stage are monitored in order to stop the computation once convergence is achieved.

### 2.6.3 Representation of Slater determinants

In order to represent Slater determinants as strings of 0s and 1s, we group together the spin-orbital pairs corresponding to the same orbital but having the opposite spins ↑↓. The leftmost pair in the string is attributed to the impurity and the further pairs correspond to the bath sites with growing energies from left to right. The content of a determinantal string is illustrated as

$$
\underbrace{\uparrow\downarrow}_{\text{imp.}} \; \underbrace{\overbrace{\uparrow\downarrow \ldots \uparrow\downarrow}^{\varepsilon_b<0} \; \overbrace{\uparrow\downarrow}^{\varepsilon_b=0} \; \overbrace{\uparrow\downarrow \ldots \uparrow\downarrow}^{\varepsilon_b>0}}_{\text{bath}} \tag{9}
$$

As the starting basis for our iterations we choose the two degenerate determinants with $S_z = 0$ and the lowest net one-particle energy in the zero-hybridization limit:

$$
10 \; 11 \ldots 11 \; 01 \; 00 \ldots 00 \,, \tag{10}
$$

$$
01 \; 11 \ldots 11 \; 10 \; 00 \ldots 00 \,. \tag{11}
$$

For given $N_{\text{bath}}$, Python strings for these determinants can be build using the function

```python
def build_start_dets(N_bath):
    det1 = "01"  + "1" * (N_bath - 1) + "10" + "0" * (N_bath - 1)
    det2 = "10"  + "1" * (N_bath - 1) + "01" + "0" * (N_bath - 1)
    return det1, det2
```

### 2.6.4 Starting Basis object

The impurity on-site interaction strength is in all examples $U = 10$. For the start, we stick to the simplest non-degenerate case of $N_{\text{bath}} = 3$ bath sites and increase later $N_{\text{bath}}$ for more advanced demonstrations.

```python
U = 10
N_bath = 3
e_bath, V_bath = build_bath(N_bath)
start_dets = build_start_dets(N_bath)
```

We follow the standard procedures described in the previous sections to construct necessary objects of the SOLAX classes. In particular, a `Basis` object containing the two starting Slater determinants (10, 11) is created as

```python
basis_start = sx.Basis(start_dets)
print(basis_start)
```

```
01111000
10110100
```

### 2.6.5 Operator object for Hamiltonian

Here we encode the Hamiltonian in parts represented by Eqs. (5)—(8). Note that for demonstration purposes, we switch the NumPy module to the regime in which up to 3 digits after the decimal point are printed (this does not influence the computational precision).

**Impurity term.** We represent the impurity on-site interaction operator as

$$\hat{H}_{\text{imp}} = \underbrace{U\,\hat{a}^{\dagger}_{\text{imp},\uparrow}\,\hat{a}_{\text{imp},\uparrow}\hat{a}^{\dagger}_{\text{imp},\downarrow}\,\hat{a}_{\text{imp},\downarrow}}_{\hat{H}^{(2)}_{\text{imp}}} - \underbrace{\frac{U}{2}\left(\hat{a}^{\dagger}_{\text{imp},\uparrow}\,\hat{a}_{\text{imp},\uparrow} + \hat{a}^{\dagger}_{\text{imp},\downarrow}\,\hat{a}_{\text{imp},\downarrow}\right)}_{\hat{H}^{(1)}_{\text{imp}}} + \frac{U}{4} \qquad (12)$$

and build up our `Operator` object term by term:

```python
H_imp2 = sx.Operator(
    (1, 0, 1, 0),
    np.array([
        [0, 0, 1, 1]
    ]),
    np.array([U])
)

H_imp1 = sx.Operator(
    (1, 0),
    np.array([
        [0, 0],
        [1, 1]
    ]),
    np.array([-U / 2, -U / 2])
)

H_imp = H_imp2 + H_imp1 + U / 4
print(H_imp)
```

```
Operator({
    (1, 0): OperatorTerm(
        daggers=(1, 0),
        posits=array([[0, 0],
               [1, 1]]),
        coeffs=array([-5., -5.])
    ),
    (1, 0, 1, 0): OperatorTerm(
        daggers=(1, 0, 1, 0),
        posits=array([[0, 0, 1, 1]]),
        coeffs=array([10.])
    ),
    scalar: 2.5
})
```

**Bath term.** Taking into account that in each bath site both spin-orbitals have the same energy, the bath Hamiltonian term $\hat{H}_{\text{bath}}$ is encoded as

```
H_bath = sx.Operator(
    (1, 0),
    np.arange(2, 2 * N_bath + 2).repeat(2).reshape(-1, 2),
    e_bath.repeat(2)
)
print(H_bath)
```

```
Operator({
    (1, 0): OperatorTerm(
        daggers=(1, 0),
        posits=array([[2, 2],
                [3, 3],
                [4, 4],
                [5, 5],
                [6, 6],
                [7, 7]]),
        coeffs=array([-1.414e+0, -1.414e+00, -1.225e-16, -1.225e-16,
                1.414e+00, 1.414e+00])
    )
})
```

**Hybridization term.**   Hybridization as described by the $\hat{H}_{\mathrm{hyb}}$ term takes place between spin-orbitals with the same spin. An `Operator` object for $\hat{H}_{\mathrm{hyb}}$ without h.c. is then built as

```
H_hyb_posits = np.vstack([
    np.array([0, 1] * N_bath),
    np.arange(2, 2 * N_bath + 2)
]).T

H_hyb_nohc = sx.Operator(
    (1, 0),
    H_hyb_posits,
    V_bath.repeat(2)
)
print(H_hyb_nohc)
```

```
Operator({
    (1, 0): OperatorTerm(
        daggers=(1, 0),
        posits=array([[0, 2],
                [1, 3],
                [0, 4],
                [1, 5],
                [0, 6],
                [1, 7]]),
        coeffs=array([1.581, 1.581, 2.236, 2.236, 1.581, 1.581])
    )
})
```

**Full Hamiltonian.** Finally we obtain an `Operator` object for the full SIAM Hamiltonian (which we don't print here):

```
H = H_imp + H_bath + H_hyb_nohc + H_hyb_nohc.hconj
```

### 2.6.6 Hamiltonian matrix and state energy

We can now obtain the Hamiltonian matrix on the basis of the 2 "starting" Slater determinants as

```
matrix_start = H.build_matrix(basis_start)
```

For this demonstration example, the obtained `OperatorMatrix` object can be converted to the NumPy dense format and printed:

```
matrix_dense_start = matrix_start.to_scipy().todense()
print(matrix_dense_start)
```

```
[[-5.328  0.    ]
 [ 0.    -5.328]]
```

This matrix is diagonal, and contains directly the state energy corresponding to the Hartree-Fock approximation:

```
energy_start = matrix_dense_start[0, 0]
print(energy_start)
```

```
-5.32842712474619
```

### 2.6.7 Basis extension

The state energy obtained above is the roughest approximation and must be refined. For this purpose, we extend the basis by acting on `basis_start` with the extension operator (here the Hamiltonian)

```
basis = H(basis_start)
print(len(basis))
```

```
8
```

The matrix built on this basis is not diagonal anymore:

```
matrix = H.build_matrix(basis)

matrix_dense = matrix.to_scipy().todense()
print(matrix_dense)
```

```
[[-5.328 -1.581 -2.236 -2.236 -1.581  0.     0.     0.   ]
 [-1.581  1.086  0.     0.     0.     0.     0.     0.   ]
 [-2.236  0.    -0.328  0.     0.     2.236  0.     0.   ]
 [-2.236  0.     0.    -0.328  0.     2.236  0.     0.   ]
 [-1.581  0.     0.     0.     1.086  0.     0.     0.   ]
 [ 0.     0.     2.236  2.236  0.    -5.328 -1.581 -1.581]
 [ 0.     0.     0.     0.     0.    -1.581  1.086  0.   ]
 [ 0.     0.     0.     0.     0.    -1.581  0.     1.086]]
```

Therefore, in order to obtain the state energy, the lowest eigenvalue has to be computed using the SciPy means (we use NumPy in this demonstration example):

```python
energy = np.linalg.eigvals(matrix_dense).min()

basis_size = len(basis)
print(f"Basis size = {basis_size}\tEnergy = {energy}")
```

```
Basis size = 8  Energy = -8.351171437060554
```

The iterations of basis extension and Hamiltonian matrix evaluation should be now repeated until the state energy converges. We switch now to a more advanced example with larger $N_{\mathrm{bath}}$ for demonstration of these iterations.

### 2.6.8   An example of full computation

We now consider the SIAM with $N_{\mathrm{bath}} = 21$ bath sites and iteratively evaluate the state energy using the described approach. The reconstruction of the `Basis` and `Operator` objects can be performed directly by rerunning the code above after assigning the new value to the `N_bath` variable. We omit this part and show the loop with the iterations directly.

```python
num_iterations = 4

basis = basis_start

for i in range(num_iterations):
    matrix = H.build_matrix(basis)
    energy = sp.sparse.linalg.eigsh(
        matrix.to_scipy(), k=1, which="SA"
    )[0][0]

    basis_size = len(basis)
    print(
        f"Iteration: {i+1:<8d}"
        f"Basis size = {basis_size:<12d}"
        f"Energy = {energy}"
    )

    if i < num_iterations - 1:
        basis = H(basis)
```

```
Iteration: 1        Basis size = 2       Energy = -28.463653910211487
Iteration: 2        Basis size = 44      Energy = -30.19530217404953
Iteration: 3        Basis size = 684     Energy = -31.242891311317756
Iteration: 4        Basis size = 7084    Energy = -31.70729257122757
```

To find the ground state in each iteration, we used the SciPy diagonalization routine `sp.sparse.linalg.eigsh` for Hermitian sparse matrices. We pass the Hamiltonian matrix and request the first smallest (`k=1`, `which="SA"`) eigenvalue. We note that the basis is not extended in the last iteration since the computation terminates immediately thereafter. It is seen that the energy is converging with the iterations, which should be stopped once the desired precision is achieved.

### 2.6.9    Optimization of matrix construction

In the iterations above we rebuilt the Hamiltonian matrix each time. Using the `OperatorMatrix` tools demonstrated in the previous section, it is possible to avoid re-evaluation of the matrix elements which have already been calculated in the previous iteration.

    To demonstrate this, we start from the `OperatorMatrix` object constructed in the last performed iteration. In Python, variables remain available after the loop is finished. Therefore, we access `basis` and `matrix` directly, and use different variable names corresponding to the analytical notations below:

```python
basis_small = basis
M_small = matrix
print(M_small.size)
```

```
(7084, 7084)
```

In the following, we will use the analytical notation $M_{\text{small}}$ corresponding to the `OperatorMatrix` object `M_small`. Now we extend the basis another time and build the Hamiltonian matrix $M_{\text{big}}^{\text{direct}}$ directly on the resulting basis (thus computing all matrix elements from scratch as before):

```python
basis_big = H(basis_small)
M_big_direct = H.build_matrix(basis_big)
print(M_big_direct.size)
```

```
(58984, 58984)
```

However, the set of Slater determinants `basis_small` is a subset of `basis_big`; this can be checked e.g. as follows:

```python
print(len(basis_small % basis_big) == 0)
```

```
True
```

Therefore, all matrix elements of $M_{\text{small}}$ enter also $M_{\text{big}}^{\text{direct}}$ allowing to avoid unnecessary re-computations. We implement here a possible scenario of such optimized matrix construction. Note, however, that $M_{\text{small}}$ is in general not a rectangular submatrix in $M_{\text{big}}^{\text{direct}}$. Instead, the matrix elements of $M_{\text{small}}$ are spread in $M_{\text{big}}^{\text{direct}}$ according to the positions

of the Slater determinants from `basis_small` in `basis_big`. In the following we construct a matrix $M_{\text{big}}$ which does contain $M_{\text{small}}$ as a true submatrix. Though $M_{\text{big}}$ and $M_{\text{big}}^{\text{direct}}$ are in general not exactly equal, they are equivalent up to permutation of the basis determinants irrelevant for our applications.

**Evaluation of the missing submatrix.** The target matrix $M_{\text{big}}$ is a Hermitian block matrix

$$M_{\text{big}} = \begin{pmatrix} M_{\text{small}} & A \\ A^\dagger & B \end{pmatrix} \tag{13}$$

with unknown blocks $A$ and $B$, where $B$ is Hermitian. Given $M_{\text{small}}$ is known, we need to additionally evaluate only the block matrix

$$C = \begin{pmatrix} A \\ B \end{pmatrix} \tag{14}$$

in order to construct $M_{\text{big}}$. The rectangular matrix $C$ is built on the following `Basis` objects:

```
basis_cols = basis_big % basis_small
basis_rows = basis_small + basis_cols
```

As mentioned in the section on the `OperatorMatrix` class, this can be achieved by passing the both `Basis` objects to the `build_matrix` method:

```
C = H.build_matrix(basis_rows, basis_cols)
print(C.size)
```

```
(58984, 51900)
```

**Constructing the matrix from its parts.** We use now the methods supported by the `OperatorMatrix` class to build up $M_{\text{big}}$ from $M_{\text{small}}$ and $C$. First of all, we bring `C` to its right place in $M_{\text{big}}$ by displacing it along the column axis, and obtain the matrix

$$C_{\text{displ}} = \begin{pmatrix} 0 & A \\ 0 & B \end{pmatrix} \tag{15}$$

```
C_displ = C.displace(0, len(basis_small))
```

Then the following sum corresponds to the block matrix

$$M_{\text{with2B}} = \begin{pmatrix} M_{\text{small}} & A \\ A^\dagger & 2B \end{pmatrix} \tag{16}$$

```
M_with2B = M_small + C_displ + C_displ.hconj
```

Now we need to subtract $B$ displaced to the proper position:

$$B_{\text{displ}} = \begin{pmatrix} 0 & 0 \\ 0 & B \end{pmatrix} \tag{17}$$

This matrix can be obtained directly from $C_{\text{displ}}$ using the `window` method:

```
left_top = (len(basis_small), len(basis_small))
right_bottom = (None, None)

B_displ = C_displ.window(left_top, right_bottom)
```

Then finally the targeted $M_{\mathrm{big}}$ matrix is obtained as

```
M_big = M_with2B - B_displ
```

**Comparison of the results.** We ensure now the equality of the state energies obtained using the direct and the optimized approaches to the matrix construction:

```
energy_big_direct = sp.sparse.linalg.eigsh(
    M_big_direct.to_scipy(), k=1, which="SA"
)[0][0]
print(energy_big_direct)
```

```
  -31.819018639483936
```

```
energy_big = sp.sparse.linalg.eigsh(
    M_big.to_scipy(), k=1, which="SA"
)[0][0]
print(energy_big)
```

```
  -31.819018639483833
```

It is evident that the obtained results agree within a very small error, which can be attributed to randomization in the SciPy eigensolver.

In applications involving particularly large Hermitian operators, such as the Hamiltonian in the computations for the $N_2$ molecule performed in Ref. [19], working directly with full operators is disadvantageous. Instead, it is possible to represent a Hermitian operator $\hat{A}$ as the sum $\hat{A} = \hat{B} + \hat{B}^\dagger$ and use only the part $\hat{B}$ for computations. In particular, this greatly simplifies the construction of the operator matrix. Once an `OperatorMatrix` is built for $\hat{B}$, adding its Hermitian conjugate creates an `OperatorMatrix` for the full operator $\hat{A}$.

## 3  Neural network support for tackling big basis sets

We have presented the basic functionality of SOLAX as a solver for fermionic quantum systems and now switch to the built-in neural network (NN) support for managing large sets of Slater determinants. In the iterative solution procedure shown for SIAM in the previous section, the energy can, in principle, be refined to any desired accuracy. In practice, however, the described basis extension approach leads to exponential basis growth and quickly becomes computationally infeasible. Here, we demonstrate the NN-based tools available in SOLAX to control basis growth and converge the results with less computational resources.

To this end, we follow the algorithm developed in Ref. [16] for managing exponentially growing bases as exemplified for SIAM. We stress that the NN support in SOLAX is not a simple function containing the entire algorithm from Ref. [16]. Instead, we provide building blocks that can be conveniently tuned and adapted by the users to their specific research problems. We show how the algorithm from Ref. [16] can be reproduced using these building blocks.

## 3.1 Algorithm description

The core idea of the algorithm from Ref. [16] is schematically illustrated in Fig. 3. Once a basis extension yields too many new Slater determinants, not all of these new "candidates" are included in the calculation. Instead the user decides which fraction $\alpha$ of *the most important* determinants is to be included. The importance of a determinant is measured by its weight, i.e. the magnitude of its expansion coefficient in the quantum state.

To fulfill the user's request, the following steps are performed. First, as illustrated in panel (A) of Fig. 3, a random selection is drawn from the set of candidate determinants and added to the existing basis used in the computation (the existing basis is not shown in Fig. 3). A diagonalization is then performed, providing the expansion coefficients for the randomly added determinants. Note that the size of the random selection is determined by the user.

Next, as illustrated in panel (B), a cutoff is chosen to divide the random selection into two classes: determinants with weights above the cutoff and those below it. We refer to these classes as "important" and "unimportant", respectively. Based on the known weights of the randomly selected determinants, the cutoff is automatically adjusted such that the important class comprises a fraction $\alpha$ of the random selection. We assume that this cutoff also divides the full set of candidates in the same proportion. Since the coefficient of each determinant may change upon the inclusion of other determinants, this assumption can only be approximately fulfilled in practice. It remains unknown *which* determinants outside the random selection belong to each class, and this is where a NN is useful.

At this stage, as shown in panel (C), a NN classifier is employed to categorize the remaining determinants into the importance classes. The random selection serves as the training data. For each determinant in the training set, the spin-orbital occupations are used as input features for the NN, while the determinant class serves as the "correct answer". Once trained, the NN is applied to the rest of the candidates to predict their importance class. This process sorts out the entire set of candidates, allowing only the important ones to be retained for diagonalization. We focus here on demonstrating this core procedure. Using the presented SOLAX tools, the algorithm can be further modified (e.g., in Ref. [16], some determinants outside the set of candidates were also used for NN training).

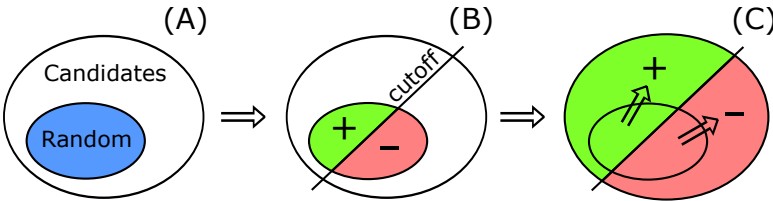

Figure 3: Schematic illustration of the method developed in Ref. [16]. Only the core part of the algorithm is shown. See text for details.

In SOLAX, we implemented a custom machine learning package based on the FLAX library [29] from the JAX ecosystem. This package is primarily intended for development purposes and is not exposed to the user at the general SOLAX interface level. We employed these machine learning tools to provide the necessary functionality for implementing algorithms such as the one described. This functionality is encapsulated in two classes available at the SOLAX interface level: `BasisClassifier` and `BigBasisManager`. The former encapsulates a NN setting, while the latter contains methods that reflect parts of the described algorithm. For demonstration, we continue with the SIAM example computation of Section 2.6.

## 3.2  BasisClassifier

We now turn to a discussion of the `BasisClassifier` class, which integrates a NN and an optimizer to facilitate necessary machine learning operations. Since `BasisClassifier` objects are not used as standalone tools but are instead passed to a `BigBasisManager` and employed there in an automated manner, we focus here only on the aspects relevant to the described algorithm.

The user is expected to create a NN architecture using FLAX building blocks, while an optimizer is selected from the Optax package available alongside JAX. First, we make the necessary imports:

```python
from flax import linen as nn
import optax
```

Here, the `flax.linen` API is imported via the conventional name `nn`. A NN architecture can be now defined using the standard FLAX approach, i.e. via writing a Python function describing how a single data entry propagates towards the classification output. In this function, the FLAX building blocks as well as general JAX transformations can be used. We implement here the convolutional architecture employed in the work [16], which was at the time implemented based on the TensorFlow library [17].

```python
def nn_call_on_bits(x):
    x = x.reshape(-1, 2)
    x = nn.Conv(features=64, kernel_size=(2,), padding="valid")(x)
    x = nn.relu(x)
    x = nn.Conv(features=4, kernel_size=(1,), padding="valid")(x)
    x = nn.relu(x)
    x = x.reshape(-1)

    x = nn.Dense(features=dense_size)(x)
    x = nn.relu(x)
    x = nn.Dense(features=dense_size//2)(x)
    x = nn.relu(x)
    x = nn.Dense(features=dense_size//4)(x)
    x = nn.relu(x)
    x = nn.Dense(features=2)(x)
    return x
```

In this function, the input `x` is an array of bits (0 and 1) representing the spin-orbital populations of the incoming Slater determinant. The alternating ↑ and ↓ bits in the

input are separated into two channels using `reshape` and processed by two convolutional layers with the ReLU activation. The output of the convolutional block is flattened using `reshape(-1)` and passed to the dense block, which ends with two neurons containing the classification logits. It is important to note that in the code that will eventually call this function, the output logits are automatically passed through the softmax activation function to convert them into probabilities that sum to 1. Therefore, there is no need to explicitly apply the softmax function within the NN definition.

In the function body above, the size of the layers in the dense block is determined by the global variable `dense_size`. We initialize it as follows:

```
dense_size = int(7 * np.sqrt(2 * N_bath + 2))
print(dense_size)
```

```
46
```

Here we provided the generic expression for arbitrary $N_{\text{bath}}$ used in Ref. [16]. An object of `BasisClassifier` is now instantiated as

```
classifier = sx.BasisClassifier(nn_call_on_bits)
```

We note that so far no NN has been actually initialized in memory. The latter needs to be performed explicitly using the `initialize` method which requires the following additional arguments: (1) an input `Basis` prototype, (2) an optimizer, and (3) a JAX random key.

The input `Basis` prototype required by the `BasisClassifier initialize` method is needed to find out the size of the input `x` of the function defining the NN. This determines also the concrete structure of the NN as created in memory. We use here the variable `basis_start` from the SIAM example in Section 2.6. We choose the standard Adam optimizer [32] which can be created using Optax as

```
optimizer = optax.adam(learning_rate=0.005)
```

### 3.2.1 RandomKeys

In JAX, randomization is performed in a deterministic and reproducible manner. Each function generating random numbers receives a JAX random key which determines the randomization. These keys can be created from each other, whereas the very first one is created from an integer seed. We wrapped this mechanism in a convenience class `RandomKeys` which implements the Python iterator interface. A `RandomKeys` instance is created as

```
rand_keys = sx.RandomKeys(seed=1234)
```

where `seed` is a keyword-only argument. Now, each time when a new JAX random key needs to be generated, the Python `next` keyword can be used on `rand_keys`:

```
key_for_nn = next(rand_keys)
```

The obtained key is passed to the `initialize` method in order to randomly initialize the NN weights.

Using the needed components, we initialize the NN in memory as

```
classifier.initialize(key_for_nn, basis_start, optimizer)
```

Note that once initialized, the NN summary can be printed out using the `print_summary` method (we skip this step here due to the output size). This exhausts the knowledge about the `BasisClassifier` class needed to proceed with implementation of the sketched algorithm using `BigBasisManager`.

## 3.3  BigBasisManager

The central SOLAX tool for implementing algorithms as the one described above, is the `BigBasisManager` class. Its objects are created from:

- a set of candidate Slater determinants represented as an object of the SOLAX class `Basis`;

- a NN setting represented as an object of the just shown SOLAX class `BasisClassifier`.

In order to demonstrate the functions provided with `BigBasisManager`, we turn back to the SIAM example with $N_{\text{bath}} = 21$ bath sites considered in Section 2.6.8. In the iterative solution shown there, we achieved the basis size of 7084 Slater determinants. Another extension yields

```
basis_small = basis

basis_big = H(basis_small)
print(len(basis_big))
```

```
58984
```

Here we stick to the notations adopted also in Section 2.6.9. The obtained basis poses no computational challenge and was tackled directly in Section 2.6.9. Still, for demonstration, we sort it out using the NN-supported algorithm and compare the resulting state energy with the one obtained directly. The set of new "candidates" generated at the last extension step is obtained as

```
candidates = basis_big % basis_small
print(len(candidates))
```

```
51900
```

Using also the `classifier` object built above, we create now a `BigBasisManager` instance:

```
bbm = sx.BigBasisManager(candidates, classifier)
```

Instead of including all 51900 candidates in the computation, we follow Ref. [16] and target inclusion of the following number of *the most important* determinants:

```
target_num = int(np.sqrt(len(basis_big)) * 50)
print(target_num)
```

```
12143
```

We provided here the generic expression from Ref. [16] based on the length of the `basis_big` set. In the following we implement the sketched NN-based approach with the help of the created `bbm` object.

### 3.3.1 Random selection

We follow Ref. [16] and choose the size of the random selection shown in Fig. 3(A) as follows:

```python
random_num = int(target_num / 1.5)
print(random_num)
```

```
8095
```

The selection can be drawn from `candidates` using the `bbm` object via the `sample_subbasis` method:

```python
random_sel = bbm.sample_subbasis(next(rand_keys), random_num)
```

Here we generated another JAX random key from the `rand_keys` object and passed it directly to the `sample_subbasis` function. The returned `random_sel` object is of type `Basis`. In order to obtain the weights for the picked determinants as required by the algorithm, we add the random selection on top of the old basis:

```python
basis_diag = basis_small + random_sel
print(len(basis_diag))
```

```
15179
```

and perform diagonalization as shown in Section 2.6.8:

```python
matrix = H.build_matrix(basis_diag)
result = sp.sparse.linalg.eigsh(matrix.to_scipy(), k=1, which="SA")

energy = result[0][0]
print(f"Intermediate energy:\t{energy}")
```

```
Intermediate energy: -31.720920015599123
```

Finally, we use the obtained eigenvector to create a `State`:

```python
eigenvec = result[1][:, 0]
state_diag = sx.State(basis_diag, eigenvec)
```

from which we strip `basis_small` obtaining the randomly selected determinants together with their coefficients encapsulated in a `State` object:

```
state_train = state_diag % basis_small
```

Note that the latter object serves only for convenient storing the random selection coefficient and does not actually represent the searched quantum state. The variable name `state_train` reflects that this `State` instance contains the data which will be used for the NN training.

As expected, the intermediate energy obtained above lies between the energy on `basis_small` computed in Section 2.6.8 and the energy on `basis_big` from Section 2.6.9. The corresponding separations are approx. 0.0136 and 0.0981. In this way, inclusion of the random selection in the computation does not considerably promote the energy towards the value on the larger basis. In contrast, as we will see in the following sections, inclusion of NN-selected determinants pushes the energy close to the value on `basis_big`.

### 3.3.2 Deriving the cutoff

We switch now to splitting the random selection with a cutoff as discussed in Section 3.1 and illustrated in Fig. 3(B). The cutoff can be obtained using the `BigBasisManager` method `derive_abs_coeff_cut`:

```
abs_coeff_cut = bbm.derive_abs_coeff_cut(target_num, state_train)
print(f"Cutoff:\t{abs_coeff_cut}")
```

```
Cutoff: 0.00020762079204639022
```

This cutoff splits the random selection into two importance classes of determinants with larger and smaller weights. We point out again that under weight we understand here the absolute value of the determinant coefficient (not its square). The "important" class as represented by a `State` object is obtained by chopping

```
state_train_impt = state_train.chop(abs_coeff_cut)
print(len(state_train_impt))
```

```
1893
```

The cutoff is automatically chosen such that the fraction of the important class in the random selection

```
print(len(state_train_impt) / len(state_train))
```

```
0.23384805435453984
```

equals the ratio of the targeted number of the most important determinants to be included to the full number of the candidates

```
print(target_num / len(candidates))
```

```
0.23396917148362234
```

We assume that the derived cutoff `abs_coeff_cut` splits also the rest of the candidates approximately in the same proportion.

### 3.3.3   Using the neural network

In the following, we employ the NN in order to distribute the determinants outside the random selection into the importance classes as illustrated in Fig. 3(C). The NN is already created, initialized and incorporated in the `bbm` object of the `BigBasisManager` class. The latter provides high-level functions for the NN usage in the considered context. We train the NN similarly to Ref. [16] as follows:

```python
early_stopped = bbm.train_classifier(
    next(rand_keys),
    state_train,
    abs_coeff_cut,
    batch_size=256,
    epochs=200,
    early_stop=True,
    early_stop_params={"patience": 3}
)
```

```
Started: accuracy=2.472703e-01
Epoch 0: accuracy=8.015612e-01
Epoch 1: accuracy=8.385798e-01
Epoch 2: accuracy=8.525778e-01
Epoch 3: accuracy=8.595095e-01
Epoch 4: accuracy=8.705761e-01
Epoch 5: accuracy=8.619702e-01
Epoch 6: accuracy=9.055037e-01
Epoch 7: accuracy=8.985720e-01
Epoch 8: accuracy=9.237778e-01
Epoch 9: accuracy=9.326122e-01
Epoch 10: accuracy=9.412651e-01
Epoch 11: accuracy=9.312204e-01
Epoch 12: accuracy=9.491246e-01
Epoch 13: accuracy=9.461933e-01
Epoch 14: accuracy=9.596332e-01
Epoch 15: accuracy=9.524729e-01
Epoch 16: accuracy=9.450301e-01
Epoch 17: accuracy=9.506576e-01
Epoch 18: accuracy=9.534478e-01
```

Here, a new JAX random key is generated from `rand_keys` and passed directly to the `train_classifier` call. It is needed for reshuffling to avoid any ordering bias in the training data. The latter are provided as the `state_train` object of the `State` class. Whereas `state_train` contains only the Slater determinants with their coefficients, the `train_classifier` method takes care of converting this to classification data based on the cutoff `abs_coeff_cut` also provided to the method.

    The rest of the parameters are keyword-only arguments controlling the NN training, which is performed in batches of size 256 until the best performance is achieved and then early-stopped. After the epoch in which the NN failed to improve, we give it a chance to try 3 more times by indicating `patience=3`. The early stopping parameters passed as the `early_stop_params` dictionary are forwarded directly to the underlying `EarlyStopping` FLAX class and can be looked up in the FLAX documentation [29]. Note that the NN is

reset to its best state achieved in the training process. The output value `early_stopped` is `True` if the training was early-stopped, and `False` if it went through all 200 epochs (provided with the `epochs` argument) without reaching the best performance.

The printed NN training information contains the NN classification accuracy evaluated before the training and after each epoch on a data part held out from the training set. The fraction of the data used for the performance evaluation can be controlled via the keyword-only argument `val_frac` of the `train_classifier` method (the default value is `val_frac=0.2`). We note that if an NVIDIA GPU is available and a GPU-capable version of JAX is installed on the machine, it will be automatically used for the NN training usually leading to a significant speedup.

The trained NN can be now used for classification of all candidates:

```python
nn_selected = bbm.predict_impt_subbasis(batch_size=256)
nn_selected = nn_selected % state_train.basis
print(len(nn_selected))
```

```
9834
```

The method `predict_impt_subbasis` returns a `Basis` object `nn_selected` containing the Slater determinants classified by the NN as important. Note that the NN is applied here to the full set of candidates including the training subset present in the `State` object `state_train`. Therefore, we strip the latter from `nn_selected`. The NN prediction operation runs automatically on a GPU if available. In this case, the keyword-only `batch_size` argument can be passed to the `predict_impt_subbasis` method for controlling the GPU memory usage.

The final subset of candidates to be included in the computation as the result of the NN-supported procedure is build as

```python
basis_impt = nn_selected + state_train_impt.basis
print(len(basis_impt))
print(abs(len(basis_impt) - target_num) / target_num)
```

```
11727
0.034258420489170716
```

Here we printed the size of the obtained important subset and its relative deviation from the targeted size `target_num`. We see that the user's request has been satisfied in this demonstration example.

### 3.3.4 Checking and processing of the results

We construct now the full basis and evaluate the state energy:

```python
basis = basis_small + basis_impt

matrix = H.build_matrix(basis)
result = sp.sparse.linalg.eigsh(matrix.to_scipy(), k=1, which="SA")

energy = result[0][0]
print(f"Basis:\t{len(basis)}")
print(f"Energy:\t{energy}")
```

```
Basis: 18811
Energy: -31.817043901573747
```

The obtained energy is separated from the energy computed on `basis_small` in Section 2.6.8 and the energy on `basis_big` from Section 2.6.9 by 0.1100 and 0.0017, respectively. In this way, by using the NN support provided in SOLAX we were able to almost reach the same state energy on 19462 Slater determinants instead of 58984.

We use now the obtained eigenvector to check how many determinants out of those suggested by the NN indeed possess weights higher that `abs_coeff_cut`. First we construct a `State` object corresponding to the NN suggestion:

```
eigenvec = result[1][:, 0]
state = sx.State(basis, eigenvec)

nn_selected_state = state % basis_small % state_train.basis
print(nn_selected_state.basis == nn_selected)
```

```
True
```

The `State` instance `nn_selected_state` contains a basis set equal to `nn_selected` and additionally the evaluated coefficients. Now we chop off the misclassified determinants:

```
nn_selected_right = nn_selected_state.chop(abs_coeff_cut).basis
print(len(nn_selected_right))
print(len(nn_selected_right) / len(nn_selected))
```

```
8648
0.8793980069147854
```

The printed fraction of the Slater determinants classified correctly is smaller that the NN accuracy achieved in the training procedure. We attribute it to the drift of the determinant coefficients upon the inclusion in the computation of other determinants.

If the obtained basis is involved in further computations (e.g. as the starting point of the next NN-supported iteration), it is advantageous to exclude from it the misclassified determinants:

```
nn_selected_wrong = nn_selected % nn_selected_right
basis_final = basis % nn_selected_wrong
print(len(basis_final))
```

```
17625
```

If needed, the energy and the coefficients can be evaluated on `basis_final` via the corresponding Hamiltonian matrix. Note, however, that it is inefficient to compute the latter from scratch using the `H.build_matrix` call. Instead, as demonstrated in the section on the `OperatorMatrix` class, the `shrink_basis` method of the `matrix` object can be used to extract the matrix on `basis_final` directly from the `OperatorMatrix` instance `matrix` built on `basis`. This shortcut is possible since `basis_final` represents a subset of `basis`, and therefore all the needed matrix elements have been already evaluated.

We point out that each `BigBasisManager` instance is bound to a particular `Basis` object with candidates to be sorted out. Therefore, it is necessary to create a new `BigBasisManager` object for each new big basis to be optimized (e.g. if further NN-supported iterations follow). At the same time, the same `BasisClassifier` object can be reused as a component in different `BigBasisManager` instances transferring in this way the NN experience from case to case.

Summarizing, using the NN support tools provided in SOLAX, we could reach the same accuracy level for the SIAM ground state energy on a much smaller basis. Further iterations of the described algorithm would profit strongly from the reduced basis size as the starting point. In Ref. [16], this algorithm was implemented using the TensorFlow library [17] and the Quanty code [18] for working with fermionic quantum systems. It was applied to SIAM with up to $N_{\text{bath}} = 299$ bath sites in order to sort out many millions of Slater determinants, and helped to reduce the necessary basis set sizes by orders of magnitude. As implemented in SOLAX, this approach was applied for the first time in Ref. [19] for computations of the ground state of the $N_2$ molecule.

# 4 Saving/loading SOLAX objects and reproducing computations

In this section, we focus on the crucial functionality of saving and loading objects of the SOLAX classes. This feature enables checkpointing computations and restoring them in a flexible and efficient way. SOLAX provides a convenient and unified mechanism for this purpose through the global functions `save` and `load`, which are compatible with objects of the `Basis`, `State`, `OperatorTerm`, `Operator`, `OperatorMatrix`, and `RandomKeys` classes. It is important to note that `RandomKeys` objects are saved in their current iteration state, rather than JAX random keys generated by them. The `BasisClassifier` class employs its own approach, which is based on the Orbax library [30] from the JAX ecosystem.

## 4.1 Standard mechanism

We start from the standard mechanism based on the `save` and `load` functions. Using this approach, it is possible to save and load objects of the SOLAX classes listed above. For instance, we can save the `basis_final` object containing the basis set of Slater determinants built using the NN-supported algorithm in the previous section:

```
sx.save(basis_final, "solax_basis_")
```

and load it again as

```
basis_loaded = sx.load("solax_basis_")
print(basis_loaded == basis_final)
```

```
True
```

Here we ensured that the loaded `Basis` object is indeed equal to the saved one. The string `"solax_basis_"` indicates the name of the directory we save to and load from.

**Important note!** The directory is created prior to saving. If the directory already exists, it will be first erased together with its current content. Note that this applies also to the `BasisClassifier` class which has its own saving/loading mechanism (see below).

Instead of saving and loading standalone SOLAX objects, it is possible to first bundle many objects in one Python dictionary. The user provides here a string label to each object as the key in a key-value pair, whereas the object itself is the value. For instance, for the objects `basis_final` and `H` from the previous section we use the string labels `"basis_from_nn"` and `"hamiltonian"`, respectively:

```python
dict_to_save = dict(
    basis_from_nn=basis_final,
    hamiltonian=H
)

sx.save(dict_to_save, "solax_basis_ham_")
```

```python
loaded_dict = sx.load("solax_basis_ham_")

for key, value in loaded_dict.items():
    print(f"{key} has type {type(value).__name__}")
```

```
basis_from_nn has type Basis
hamiltonian has type Operator
```

We printed here only the types of the loaded objects. The used key strings are employed in internal addressing of the saved objects and must satisfy the following rule: A key string is valid if it *could be* used as a Python variable name. Therefore, we recommend the user to create dictionaries for saving using the `dict` constructor as above instead of the Python literal `{...}`. In this case, the formulated rule will be satisfied automatically, since the key strings actually *are* used as variable names.

It is possible to nest such dictionaries as above, and add variables of many standard Python types or NumPy arrays. This allows to perform a unified saving in a comprehensive way. For example, the following dictionary can be saved using directly the `save` function:

```
dict_to_save = dict(
    info="This computation is a demonstration of SOLAX",
    params=dict(
        N_bath=N_bath,
        U_impurity=U
    ),
    basis_from_nn=basis_final,
    last_epochs=dict(
        epochs=np.array([14, 15, 16, 17, 18]),
        accuracies=np.array([9.596332e-01, 9.524729e-01, 9.450301e-01,
                             9.506576e-01, 9.534478e-01])
    ),
    random_keys_after=rand_keys
)

sx.save(dict_to_save, "solax_big_save_")
```

At the technical level, the `save` and `load` functions process SOLAX objects in the following way:

- the standard Python types are converted to the JSON format [33] which is saved and loaded using the standard Python `json` module;

- the underlying NumPy arrays are saved and loaded using the NumPy means (the `RandomKeys` class needs additionally conversion of JAX arrays to and from NumPy).

We deliberately refrained from using the well known `pickle` module from the Python standard library, since it has been shown to have safety flaws [34].

## 4.2 Saving/loading BasisClassifier objects

The `BasisClassifier` class does not follow the standard saving/loading mechanism described above. Instead, it implements its own methods `save_state` and `load_state` based on the Orbax library [30]. Saving is performed in a straightforward way:

```
classifier.save_state("solax_nn_")
```

**Important note!**   As in the case of the global `save` function, the shown method creates the directory before the `BasisClassifier` object is saved there. If the directory already exists, it will be first erased together with its current content.

In order to reconstruct the saved `BasisClassifier` object, the user needs first to create and initialize a new `BasisClassifier` instance following the procedure described in Section 3.2:

```
loaded_nn = sx.BasisClassifier(nn_call_on_bits)

fake_key = sx.RandomKeys.fake_key()
loaded_nn.initialize(fake_key, basis_start, optimizer)
```

Here we reused the `nn_call_on_bits` function defining the NN architecture, `basis_start` as a prototype `Basis` instance, and the `optimizer` object. Instead of using a properly

constructed JAX random key, we obtained a "fake" key directly from the `RandomKeys` class (i.e. without creating an instance and making an iteration step). Such fake keys should not be applied in true randomization, but can be used e.g. to initialize a NN whose neuron weights and biases will be anyway overwritten by loading. The saved NN state can be now restored as

```
loaded_nn.load_state("solax_nn_")
```

The presented tools are complete to provide the user with the possibility to conveniently and efficiently save and load SOLAX computations.

### 4.3   A note on randomization under GPU acceleration

The JAX library and its derivatives like FLAX used here for the NN implementation, treat randomness in a deterministic way based on random keys. However, in GPU-accelerated computations some non-deterministic effects have been observed, see e.g. the discussion [35]. In our computations, we observed deviations of the NN training accuracies when rerunning the script from scratch. This could be avoided by adding the following lines immediately after the imports:

```
import os
os.environ['XLA_FLAGS']='--xla_gpu_deterministic_ops=true'
```

In subsequent JAX versions a different approach might be needed or the problem might be completely resolved.

## 5   Conclusions

In this work, we have demonstrated the foundational capabilities of the SOLAX library for tackling complex fermionic many-body quantum systems. The core components of SOLAX implemented as the `Basis`, `State`, `OperatorTerm`, `Operator` and `OperatorMatrix` classes, provide a robust framework for encoding and manipulating bases of Slater determinants, constructing quantum states, and handling operators within the second quantization formalism. Through a detailed application to the Single Impurity Anderson Model (SIAM), we illustrated how the iterative extension of the basis set combined with diagonalization can be efficiently implemented using SOLAX.

When the basis set grows too large to handle using available computational resources, the SOLAX built-in neural network (NN) support offers a practical solution. This approach allows for an efficient approximation of quantum states by identifying and selecting the most significant Slater determinants, thereby reducing the computational burden. The presented SIAM results, together with the SOLAX-based computations for the paradigmatic $N_2$ molecule performed in Ref. [19], underscore the flexibility and power of SOLAX in dealing with large basis sets. The modular integration of state-of-the-art machine learning techniques into the SOLAX framework opens up new avenues for addressing larger and more complex quantum systems that were previously beyond reach with alternative methods.

Looking forward, the development of additional toolboxes to further leverage the power of NNs for solving challenging computational problems in quantum many-body physics is planned for future versions of SOLAX.

# Acknowledgements

PB thanks Fred Baptiste for his valuable Python lessons. We gratefully acknowledge the group of Hannes Jónsson at the University of Iceland for their collaboration on the $N_2$ molecule [19] which was the first application of SOLAX. The authors further gratefully acknowledge the scientific support and HPC resources provided by the Erlangen National High Performance Computing Center (NHR@FAU) of the Friedrich-Alexander-Universität Erlangen-Nürnberg (FAU). PB gratefully acknowledges the ARTEMIS funding via the QuantERA program of the European Union.

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
