# Peer review of "SOLAX: A Python solver for fermionic quantum systems with neural network support"

_SciPost Physics Codebases, doi:SciPost Phys. Codebases 51 (2025) , SciPost Phys. Codebases 51-r1.0 (2025)_

## Round 1 · Referee Report · Anonymous (Referee 1) · 2024-10-29

Strengths

1 - Easy to install
2 - In particular the solver is well documented and its usage is straightforward
3 - The userguide is easy to read and provides examples to get started
4 - The library makes use of JAX to allow for Neural Network (NN) support

Weaknesses

1 - The NN support is providing very little background and uses machine learning jargon which needs further explanation
2 - In a similar vain, the example calculation (solving the Single Impurity Anderson Model, SIAM) needs more context

Report

This manuscript provides an introduction to SOLAX, a python library for solving fermionic quantum systems (in particular impurity models) with neural network support. In my opinion, this manuscript and the corresponding code are well suited for a publication in SciPost Physics Codebases after some corrections and changes as detailed below. In particular, I think that the acceptance criteria for a Codebases article are met if further benchmarking tests are provided with the source code. However, I do not think that it should be published as a SciPost Physics article, the category under which it has been (perhaps by mistake) submitted.
The corresponding program code can be installed rather easily, but the authors might still want to add a few remarks concerning the required versions of the solax and orbax packages (see ‘requested changes’ below).

The userguide is divided into three main parts: A description of the solver, the neural network support, and the saving and loading of SOLAX objects.
The largest part of the userguide, chapter 2, focusses on the functionalities of the solver, in particular on the Basis, State, OperatorTerm, Operator, and OperatorMatrix classes. The different functionalities of these classes are nicely explained in a pedagogical manner that does not require specialized knowledge and is very well suited for users to get started with the SOLAX library. Ample example code snippets illustrate the functionalities and jupyter notebooks are provided for the three main chapters of the text.
Part three is, however, hard to follow through for a non-expert. Especially readers with a background in solid state physics or quantum chemistry without prior detailed knowledge of machine learning and deep learning concepts might experience difficulties. This is unfortunate since it could be easily the target readership of this article. I think the paper would benefit largely from a revision which aims at better explaining the key concepts and providing enough information to understand this chapter without having to read the NN books mentioned in the introduction or the authors’ paper on solving the SIAM, Ref. [16]. I understand that certain basic concepts using NN can be expected from a user of SOLAX, but a more pedagogical approach in chapter 3 as detailed in the requested changes below would still be desirable.

Concerning the pros and cons of the presented solver, it is clearly stated in the conclusions that the main advantage of SOLAX is to allow treating large bath sizes of up to 299 bath sites in the context of the SIAM. However, the authors should also comment on the limitations of the solver and mention how much computation time is spent on the training of the NN in the two mentioned cases (SIAM, N$_2$ molecule) . Furthermore it would be useful for a reader to know in which situations a large number of sites is actually needed for the presented (single-orbital) SIAM.

Requested changes

1 - In the installation section, the authors specify which versions of the required python libraries have been used to run the example scripts shown in the userguide. This is perfectly fine. However, it might be useful to point out a few delicate points concerning the flax and orbax libraries. In particular when using the Anaconda ecosystem, the provided library versions are not sufficient to run the jupyter notebooks correctly. Also, it might be worth pointing out to opt for the most recent orbax-checkpoint version. 2 - In section 2.2.3 A, please specify how the chop method handles complex coefficients of the State objects. 3 - As a suggestion to improve the readability of equations (1), (2), etc., adding the corresponding formulas in human readable notation might help, i.e. $a^{\dagger}_{1,\uparrow}$ instead of $a_{0}^{\dagger}$. In the same spirit, I would suggest to add some more physical context whenever it is possible, e.g. on page 16, section 2.3.4, where a singlet state is encoded. 4 - On page 17 experimental features are mentioned. Please say explicitly which features are tested and which are experimental in the current version of the code! In the same spirit, it is mentioned on page 15, section 2.3.1 that there are other “manual mode” features which are not detailed in the userguide. Where are these functions documented? 5 - The example matrix on pages 23/24ff. seems to have a wrong diagonal part. At least when following the notation in the userguide (and also the jupyter notebook) I obtain for the specified basis a different potential energy due to the U term: on the diagonal (1.0, 1.25, 1.0, 1.75). The example matrix should be corrected. 6 - Concerning the code snippets in the userguide, on page 5, sec. 2.1, solax and numpy are imported. It would be consistent to import there already scipy, which is needed later for the sparse eigensolver on page 34. 7 - In Figure 1 the level scheme is slightly confusing. From this figure it is not clear that the hybridizations $V_b$ and energies $\epsilon_b$ are different for the different bath sites. In particular the “energy arrow” needs to be updated in order to avoid the impression that all bath levels are situated at the same energy. 8 - Page 29: Please add a brief motivation and introduce the density of states (semi-circular) and the hybridization (cosine) used in the example calculation. A more compact version of the intro in Sec. III of Ref. 16 around eq. (16) might be a good starting point. Page 30: Similarly, briefly motivate and explain the choice of GS via two Slater determinants (compare to the passage around eq. (17) of Ref.16). 9 - Concerning chapter 3, I think in particular the function nn_call_on_bits needs better explanation (section 3.2, page 39f.). Please avoid abbreviations like ReLU (rectified linear unit), explain the dense block, comment on why there is no padding in the convolutional layer, etc. Also, a figure which explains the architecture of the NN such as Fig. 2 of Ref. 16 might help. 10 - In chapter 3, it would be good to clearly separate parameters with formal justification from those which have been optimized, fine-tuned or chosen based on “machine learning experience”. In particular the choices of the following parameters are not clear: (a) dense_size – explain the generic expression used here ; (b) It is not clear on which basis the number of most important determinants has been determined (target_num on page 41). This needs to be better explained. (c) Why taking 2/3 of the target set size for the random set used for training? Is this factor of 2/3 fine-tuned or can it be justified in a formal way? 11 - Two minor corrections: - page 36, center “[…] passing both Basis objects[…]” (without 'the') - page 43, first sentence: “[...]for conveniently storing[...]”

Recommendation

Accept in alternative Journal (see Report)

---

## Round 1 · Referee Report · Anonymous (Referee 2) · 2024-11-19

Strengths

1- The codebase addresses a clear need with well-justified design decision and modular implementation. 2- The code is based on existing packages and frameworks and integrates them well. The installation procedure seems quite simple. 3- The code is compatible "out-of-the-box" with existing NVIDIA GPUs easily enabling parallelization. 4- The provided documentation is clear and easy to follow, the detailed examples are appreciated.

Weaknesses

1- I find that the paper could be more self-contained by not relying so much on some of the cited works in which methods were introduced and providing more context. 2- The paper would benefit from more discussion of limitations and future directions/updates.

Report

This paper introduces the SOLAX Python package which provides utilities for working with fermionic quantum states and operators in the second quantization. It allows to construct and manipulate arbitrary basis sets, and express quantum states and operators in these bases. It leverages existing standard libraries such as Numpy, SciPy and JAX and is compatible with NVIDIA GPUs for parallelization. It also includes utilities to select basis states in large Hilbert spaces using neural networks and thus improve the efficiency of ground-state computations.

I think this codebase is a nice contribution and will likely be useful amongst practitioners, especially those interested in applications of deep learning to ground-state computations. Being a reader familiar with the machine learning side more than the condensed matter one, I found that the examples given could benefit from a bit more discussion on the physics (see requests). I would also appreciate some more on potential limitations and potential problem cases of the machine-learning basis selection method.

The article would be well-suited for publication in SciPost Physics Codebases. I would recommend its acceptance there with minor modifications. I don't think it meets the criteria for SciPost Physics since it does not present a new method or results.

Requested changes

1- Are there libraries that implement similar functionalities? If yes, what are the main differences in capabilities and design choices between the proposed libraries and alternative ones? 2- I would like to see a more detailed discussion of what the main future directions would be so that potential users have an idea of what to look for in coming releases or things they could contribute. 3- For operators, explain quickly dagger convention as there are multiple possible orderings. 4- If no batch size is specified, will the GPU be used by default? This should be clarified. 5 - "We assume that this cutoff also divides the full set of candidates in the same proportion." This seems like a strong assumption. Is it really necessary to make it? If yes, can the authors provide an argument for why this assumption is reasonable. A trivial failure case is for example when then ground state is one of the basis states. Are there other likely ones? 6- To make the paper more self-explanatory can some more motivation for the deep learning approach can be provided? Why should it be expected to work well and generalize beyond the training set? 7- What are potential limitations and potential problem cases of the machine-learning basis selection method? There should be a discussion of these. 8- I did not understand why a random instance is generated each time a random sample is needed. Some more details on this in 3.2.1 would be good. 9- Can the authors provide more discussion about the relevance of the SIAM problem? It would also be necessary to provide some short justification on the choice of Hamiltonian parameters. 10- Can the authors provide some more discussion on the basis extension method along with references? What are some strengths and limitations of this method?

The following are not requested changes, but features to consider: - It would be nice to have a "fancy" printing option for operators to display them in a more readable format (sum of creation-annihilation operators products). - Adding the possibility to label the orbitals in some meaningful way could be an interesting feature, to allow the neural network to rely on these labels. For example, explicitly what is the spin value, or angular momentum quantum number associated with a given orbital. Leveraging this information instead of just the position of the bits in the Slater determinant could help the neural network make better predictions.

Recommendation

Accept in alternative Journal (see Report)

---

## Round 2 · Author Response

Dear Editor,

We sincerely thank the Referees for their thorough review and their positive feedback. We appreciate the time and effort taken to provide constructive comments, which have helped us to improve the presentation of our SOLAX library. With this letter, we submit a revised version of the manuscript with the requested changes for its publication in SciPost Physics Codebases.

Response to Referee 1

Referee 1

This manuscript provides an introduction to SOLAX, a python library for solving fermionic quantum systems (in particular impurity models) with neural network support. In my opinion, this manuscript and the corresponding code are well suited for a publication in SciPost Physics Codebases after some corrections and changes as detailed below. In particular, I think that the acceptance criteria for a Codebases article are met if further benchmarking tests are provided with the source code. However, I do not think that it should be published as a SciPost Physics article, the category under which it has been (perhaps by mistake) submitted.

The corresponding program code can be installed rather easily, but the authors might still want to add a few remarks concerning the required versions of the solax and orbax packages (see ‘requested changes’ below).}

The userguide is divided into three main parts: A description of the solver, the neural network support, and the saving and loading of SOLAX objects. The largest part of the userguide, chapter 2, focusses on the functionalities of the solver, in particular on the Basis, State, OperatorTerm, Operator, and OperatorMatrix classes. The different functionalities of these classes are nicely explained in a pedagogical manner that does not require specialized knowledge and is very well suited for users to get started with the SOLAX library. Ample example code snippets illustrate the functionalities and jupyter notebooks are provided for the three main chapters of the text.

Part three is, however, hard to follow through for a non-expert. Especially readers with a background in solid state physics or quantum chemistry without prior detailed knowledge of machine learning and deep learning concepts might experience difficulties. This is unfortunate since it could be easily the target readership of this article. I think the paper would benefit largely from a revision which aims at better explaining the key concepts and providing enough information to understand this chapter without having to read the NN books mentioned in the introduction or the authors’ paper on solving the SIAM, Ref. [16]. I understand that certain basic concepts using NN can be expected from a user of SOLAX, but a more pedagogical approach in chapter 3 as detailed in the requested changes below would still be desirable.

Response
We thank the Referee for the overall very positive feedback and the constructive criticism. We have added a new Section 3.1 with a pedagogical introduction to the basic concepts of NNs relevant for the purposes of the present work. Additionally, we have included a new Section 2.6.1 with an introduction to the SIAM.

Referee 1

Concerning the pros and cons of the presented solver, it is clearly stated in the conclusions that the main advantage of SOLAX is to allow treating large bath sizes of up to 299 bath sites in the context of the SIAM. However, the authors should also comment on the limitations of the solver and mention how much computation time is spent on the training of the NN in the two mentioned cases (SIAM, N2molecule). Furthermore it would be useful for a reader to know in which situations a large number of sites is actually needed for the presented (single-orbital) SIAM.

Response
In order to address the limitations of SOLAX and the underlying NN-based approach from the prospective of the computational resources, we have added a new Section 3.4.5 with computational time benchmarks for the presented NN-supported algorithm. In this section, we consider computations for the SIAM as the central example in the present work. We mention also the difference in the computational benchmarks for the N$_2$ molecule, however, without going into details.

We note also that conceptually, applicability of the presented method to a particular case is itself a research question, which can be explored using the SOLAX tools e.g. as described in Section 3.4.4 "Checking and processing of the results"

Concerning the number of bath sites in the SIAM we note that the original SIAM involves a continuous bath. Hence, to accurately approximate the continuous bath, the maximally feasible number of bath sites is desirable. In the revised version of the manuscript, this point is addressed in the added Section 2.6.1 with the SIAM introduction.

Requested Changes

Referee 1

1 In the installation section, the authors specify which versions of the required python libraries have been used to run the example scripts shown in the userguide. This is perfectly fine. However, it might be useful to point out a few delicate points concerning the flax and orbax libraries. In particular when using the Anaconda ecosystem, the provided library versions are not sufficient to run the jupyter notebooks correctly. Also, it might be worth pointing out to opt for the most recent orbax-checkpoint version.

Response
We thank the Referee for pointing out these issues. Indeed, there are subtleties in the installation procedure for packages from the JAX ecosystem via conda. We refer to the installation guide on the webpage where these and other installation aspects are addressed. We added this link in the amended version of the manuscript.

We agree with the Referee that it is necessary to follow new versions of the packages used in SOLAX. However, JAX is a new ecosystem which is still under extensive development with new versions appearing frequently. For the current publication we decided to stick to the versions listed in Table I. In further SOLAX versions we plan to take into account the development progress for the dependencies.

In Section 1.1, we have added a link Ref. [1] with the JAX installation instructions (in the amended version of the manuscript Ref. [35]), and the sentence "For installing JAX, we suggest to follow the instructions on the webpage [35], where different installation aspects are addressed.".

We have also added the following sentences: "We also note that compatibility of the presented SOLAX version with newer versions of the listed packages cannot be guaranteed, especially for the libraries from the JAX ecosystem which are still under extensive development. In further SOLAX versions we plan to take into account the development progress for the dependencies."

Referee 1

2 In section 2.2.3 A, please specify how the chop method handles complex coefficients of the State objects.

Response
We have changed the formulation in the manuscript from "magnitude smaller than threshold" to "absolute value smaller than threshold". This is also true for complex coefficients.

Referee 1

3 As a suggestion to improve the readability of equations (1), (2), etc., adding the corresponding formulas in human readable notation might help, i.e. $a^\dag_{1,\uparrow}$ instead of $a^\dag_{0}$. In the same spirit, I would suggest to add some more physical context whenever it is possible, e.g. on page 16, section 2.3.4, where a singlet state is encoded.

Response
We have extended Eqs. (1)---(3) with textbook notation formulas. Additional physical context is now provided in the extended discussion of the SIAM case (Section 2.6.1).

Referee 1

On page 17 experimental features are mentioned. Please say explicitly which features are tested and which are experimental in the current version of the code!
In the same spirit, it is mentioned on page 15, section 2.3.1 that there are other “manual mode” features which are not detailed in the userguide. Where are these functions documented?

Response
We thank the Referee for this suggestion. The only experimental feature available to the user is the possibility to perform some computations on a few GPUs in parallel. In the revised version of the manuscript, we made sure that it is stressed every time this feature is mentioned. We have additionally pointed it out in Section 1.1: "The current version of SOLAX includes a possibility to perform some computations in parallel on multiple GPUs. Note, however, that this functionality is still under development, and is considered here as an experimental feature."

The "manual mode" is intended not for the end user, but for the development purposes. We have decided to exclude it completely from the revised version of the manuscript in order to keep it at the level of a general tutorial. Presentation of this and other features useful for further SOLAX development will become a part of the SOLAX documentation, which we plan to add to the SOLAX GitHub repository.

Referee 1

4 The example matrix on pages 23/24ff. seems to have a wrong diagonal part. At least when following the notation in the userguide (and also the jupyter notebook) I obtain for the specified basis a different potential energy due to the U term: on the diagonal (1.0, 1.25, 1.0, 1.75). The example matrix should be corrected.

Response
We thank the Referee for pointing out this unfortunate mistake, which is corrected in the amended version of the manuscript. We note that this occurred due to copying of an output of a different example and is not a problem in SOLAX.

Referee 1

5 Concerning the code snippets in the userguide, on page 5, sec. 2.1, solax and numpy are imported. It would be consistent to import there already scipy, which is needed later for the sparse eigensolver on page 34.

Response
We thank the Referee for pointing this out. Indeed, there should be an import of SciPy added. However, in contrast to NumPy which is fundamental for the SOLAX interface, SciPy plays a very specialized role in the communication of the user with SOLAX. Namely, a SciPy object is returned by the method OperatorMatrix.to_scipy. Note that still the user may decide to process this output using NumPy means as e.g. in Section 2.6.8, in which case there is no need to import SciPy explicitly. Only in Section 2.6.9 the module is used explicitly for the first time. This is where we added the SciPy import in the amended version of the manuscript.

Referee 1

6 In Figure 1 the level scheme is slightly confusing. From this figure it is not clear that the hybridizations $V_b$ and energies $\varepsilon_b$ are different for the different bath sites. In particular the “energy arrow” needs to be updated in order to avoid the impression that all bath levels are situated at the same energy.

Response
In the revised version of the sketch we now indicate more clearly that $b$ is an index and $V_b$ as well as $\varepsilon_b$ are different energies on the "E" axis for different bath sites.

Referee 1

7 Page 29: Please add a brief motivation and introduce the density of states (semi-circular) and the hybridization (cosine) used in the example calculation. A more compact version of the intro in Sec. III of Ref. 16 around eq. (16) might be a good starting point. Page 30: Similarly, briefly motivate and explain the choice of GS via two Slater determinants (compare to the passage around eq. (17) of Ref.16).

Response
We have revised the presentation of the SIAM and the model Hamiltonian along the lines suggested by the Referee. As explained in Section 2.6.2, the choice of the parameters which allows a direct mapping between the star and chain geometry should now be much clearer. Also the choice for our initial Slater determinants is now explained more comprehensively in Section 2.6.3.

Referee 1

8 Concerning chapter 3, I think in particular the function nn_call_on_bits needs better explanation (section 3.2, page 39f.). Please avoid abbreviations like ReLU (rectified linear unit), explain the dense block, comment on why there is no padding in the convolutional layer, etc. Also, a figure which explains the architecture of the NN such as Fig. 2 of Ref. 16 might help.

Response
In the revised version of the manuscript, we motivate the architecture of the NN in the NN introduction, explain the abbreviations there and have added a figure with the NN architecture illustrating the function nn_call_on_bits similarly to Ref. [16]. We have, however, kept the abbreviation ReLU since it is a commonly used notation, but explained its role better in the introduction to NNs.

Referee 1

9 In chapter 3, it would be good to clearly separate parameters with formal justification from those which have been optimized, fine-tuned or chosen based on “machine learning experience”. In particular the choices of the following parameters are not clear: - dense_size– explain the generic expression used here ; - It is not clear on which basis the number of most important determinants has been determined (target_num on page 41). This needs to be better explained. - Why taking 2/3 of the target set size for the random set used for training? Is this factor of 2/3 fine-tuned or can it be justified in a formal way?

Response
Optimization of these parameters poses case-by-case research questions in their own right. These functions empirically turned out to be a good choice in the test-cases SIAM and N$_2$. The tools provided in SOLAX offer users a possibility to test this empirical choice for their problem at hand or introduce their own values. We have commented this in the text as the corresponding formulas for the parameters are introduced.

Referee 1

10 Two minor corrections:
- page 36, center “[…] passing both Basis objects[…]” (without 'the')
- page 43, first sentence: “[...]for conveniently storing[...]”

Response
We thank the Referee for pointing out these typos, which have been fixed now.

Response to Referee 2

Referee 2

This paper introduces the SOLAX Python package which provides utilities for working with fermionic quantum states and operators in the second quantization. It allows to construct and manipulate arbitrary basis sets, and express quantum states and operators in these bases. It leverages existing standard libraries such as Numpy, SciPy and JAX and is compatible with NVIDIA GPUs for parallelization. It also includes utilities to select basis states in large Hilbert spaces using neural networks and thus improve the efficiency of ground-state computations.

I think this codebase is a nice contribution and will likely be useful amongst practitioners, especially those interested in applications of deep learning to ground-state computations. Being a reader familiar with the machine learning side more than the condensed matter one, I found that the examples given could benefit from a bit more discussion on the physics (see requests). I would also appreciate some more on potential limitations and potential problem cases of the machine-learning basis selection method.

The article would be well-suited for publication in SciPost Physics Codebases. I would recommend its acceptance there with minor modifications. I don't think it meets the criteria for SciPost Physics since it does not present a new method or results.

Response
We thank the Referee for the positive assessment of our work and the suggestions on possible improvements, which we address below. We fully agree with the Referee, that the manuscript should be published in SciPost Physics Codebases instead of SciPost Physics.

Requested Changes

1 Are there libraries that implement similar functionalities? If yes, what are the main differences in capabilities and design choices between the proposed libraries and alternative ones?

Response
We thank the Referee for this important question. In the emerging domain of NN-supported computations for many-body quantum systems there are not yet many codes available to the general audience. The problem is exacerbated by the fact that in the past codes for selection of basis states were written in Fortran or similar languages, whereas modern NN functionalities are provided primarily in Python. A remarkable modern example to point out is the NetKet code, since similarly to SOLAX it is based on the JAX library. However, NetKet is build around neural quantum states, i.e. NNs which approximatively encode states of quantum many-body systems, whereas SOLAX uses an NN classifier to select important basis states. To the best of our knowledge, SOLAX is the first integrated JAX-based implementation of such NN-supported approach. We have provided this information in the Introduction in the revised version of the manuscript.

2 I would like to see a more detailed discussion of what the main future directions would be so that potential users have an idea of what to look for in coming releases or things they could contribute.

Response
An extended outlook section has been added to the revised version of the paper to detail the planned future developments of the SOLAX package. This includes interfacing with existing computational codes, the implementation of predefined operators, and features such as spectral function computation and embedding schemes, emphasizing modularity and community engagement. The additions aim to provide a clear vision for the continued evolution and broad spectrum of applications of SOLAX.

3 {For operators, explain quickly dagger convention as there are multiple possible orderings.}

Response
Indeed, the same ladder operators can be ordered in different ways. The standard choice in quantum mechanics is the "normal" ordering, in which creation operators precede annihilation operators. In SOLAX, ordering is fully the choice of the user which is controlled by the argument daggers, which indicates the positions of the creation and annihilation operators. We note that in the current version of SOLAX, no transformation is performed between different ordering choices. In Section 2.3.1 in the amended version of the manuscript, we have added the sentence "We stress that in SOLAX, the ordering of ladder operators is fully the choice of the user which is controlled by the argument daggers indicating the positions of the creation and annihilation operators".

4 {If no batch size is specified, will the GPU be used by default? This should be clarified.}

Response
We believe this confusion may have arisen from the necessity to specify a batch size when using multiple GPUs. Therefore, we have revised and improved the part "Multiple GPUs" in Section 2.3.5 "GPU acceleration and batches".

5 {"We assume that this cutoff also divides the full set of candidates in the same proportion." This seems like a strong assumption. Is it really necessary to make it? If yes, can the authors provide an argument for why this assumption is reasonable. A trivial failure case is for example when then ground state is one of the basis states. Are there other likely ones?}

Response
The Referee raises an important point here and we agree with the Referee that our original statement is generally too strong as the assumption does not have to be satisfied exactly. Instead, it should be stated that the better this condition is fulfilled, the better the random selection represents the full set of candidates, and, therefore, the higher is the quality of the NN prediction for the full set based on the information extracted from the training set.

Importantly, the user does not have to check the ratios explicitly at all if the algorithm performance is monitored after each NN-supported iteration as we demonstrate in Section~3.4.4.

In order to make this clearer, we have replaced the original statement by the following sentence: "If the algorithm is well applicable to the case at hand, this cutoff divides also the full set of candidates in a similar proportion (note that the algorithm performance should be investigated for each specific case e.g. as we demonstrate in Section~3.4.4.)".

Finally, we agree that the NN-assisted selection procedure would indeed run into problems for a ground state consisting of a single determinant. However, such cases would be easily detected by monitoring the output yielded by the parts of the algorithm. In addition, using the Hamiltonian as extension operator presents another fail-safe as no new candidates would be generated in the extension step which can be used as a trigger to exit the iterative procedure.

6 {To make the paper more self-explanatory can some more motivation for the deep learning approach can be provided? Why should it be expected to work well and generalize beyond the training set?}

Response
First of all, to enhance the clarity and make the paper more self-explanatory, we have included a new Section 3.1 where we introduce and motivate the NN concepts relevant to the present work. Whereas we prefer NNs due to their scalability, flexibility and availability of powerful frameworks like JAX and TensorFlow, also other classifier types could be potentially employed in the described algorithm, as demonstrated in Refs [2, 3] for similar computations. We have added this information in the summary of the NN-supported approach in Section 3.4.4.

7 {What are potential limitations and potential problem cases of the machine-learning basis selection method? There should be a discussion of these.}

Response
We thank the Referee for raising this important point, which can be addresses in a two-fold way. From the conceptual perspective, the applicability of the described approach to a particular problem is itself a research question. With SOLAX, we provide tools to address this question. From the technical perspective, i.e. in the context of the needed computational resources, we address this question in the revised version of the manuscript by adding a new Section 3.4.5 with time benchmarks for the algorithm executed with GPU acceleration and purely on CPU. We have also added a discussion of this point at the end of the summary of the NN-supported approach in Section 3.4.4.

8 I did not understand why a random instance is generated each time a random sample is needed. Some more details on this in 3.2.1 would be good.

Response
We assume the Referee addresses here the JAX random keys that are generated each time when a new random sample needs to be generated. These random keys ensure that the randomness in the calculations is controlled, making the results reproducible. A unique random key must be used for each random sample. Reusing the same key twice in the same function leads to the same result. Even reusing the same key in different randomization functions can introduce bias in the calculations and is considered a poor practice.

9 Can the authors provide more discussion about the relevance of the SIAM problem? It would also be necessary to provide some short justification on the choice of Hamiltonian parameters.

Response
We have significantly extended the introduction to the SIAM model and provided more context to its relevance in solid state physics in the newly added Section 2.6.1. We have also extended the description of the SIAM Hamiltonian in Section 2.6.2 and explained the choice of the parameters which allow for a one-to-one mapping of the star geometry to the chain geometry.

10 Can the authors provide some more discussion on the basis extension method along with references? What are some strengths and limitations of this method?}

Response
We thank the Referee for this suggestion. In the revised version of the manuscript we have provided more information on the basis extension procedure, see Section 2.6.8. In particular, we point out that using the Hamiltonian as the extension operator has the advantage to preserve the symmetry of the determinants and therefore, we automatically generate only determinants with non-zero contribution to the ground state. This, however, can become computationally costly for more complex Hamiltonians, e.g., for atoms and molecules including inter-orbital interactions, see the discussion in the end of Section 3.4.5.

The following are not requested changes, but features to consider: - It would be nice to have a "fancy" printing option for operators to display them in a more readable format (sum of creation-annihilation operators products).

Response

We thank the Referee for this suggestion. We are currently working on making the interface more user-friendly. The interface updates will be included in future SOLAX versions.

  • Adding the possibility to label the orbitals in some meaningful way could be an interesting feature, to allow the neural network to rely on these labels. For example, explicitly what is the spin value, or angular momentum quantum number associated with a given orbital. Leveraging this information instead of just the position of the bits in the Slater determinant could help the neural network make better predictions.

Response
We thank the Referee for this suggestion. Indeed, a possibility to label orbitals might be a convenient feature for the user. However, the current NN-supported algorithm in SOLAX is independent of any orbitals labelling. Also the positions of the orbitals are not used, but only the number of electrons at each position.

Conclusion

In conclusion, we are grateful to the Referees for the comments which helped us to considerably improve our manuscript, and hope to have addressed them sufficiently for publication in SciPost Physics Codebases.

Sincerely,
Louis Thirion, Philipp Hansmann, and Pavlo Bilous

References

[1] https://jax.readthedocs.io/en/latest/installation.html.
[2] WS. Jeong, C. A. Gaggioli, and L. Gagliardi, Active Learning Configuration Interaction for Excited-State Calculations of Polycyclic Aromatic Hydrocarbons, J. Chem. Theory Comput. 17, 7518-7530 (2021).
[3] S. D. P. Flores, Chembot: A Machine Learning Approach to Selective Configuration Interaction, J. Chem. Theory Comput.17, 4028-4038 (2021).

---

## Round 2 · List of Changes

Added sections: - Introduction to neural networks (Section 3.1) - Introduction to SIAM (Section 2.6.1) - Computational time benchmarks (Section 3.4.5)

(see our reply to the Referee reports for the other changes)

---

## Editorial Decision

published